# TRUTH-VALUE JUDGMENT IN LANGUAGE MODELS: BELIEF DIRECTIONS ARE CONTEXT SENSITIVE

## ABSTRACT

Recent work has demonstrated that the latent spaces of large language models (LLMs) contain directions predictive of the truth of sentences. Multiple methods recover such directions and build probes that are described as uncovering a model's "knowledge" or "beliefs". We investigate this phenomenon, looking closely at the impact of *context* on the probes. Our experiments establish where in the LLM the probe's predictions are (most) sensitive to the presence of related sentences, and how to best characterize this kind of sensitivity. We do so by measuring different types of consistency errors that occur after probing an LLM whose inputs consist of hypotheses preceded by (negated) supporting and contradicting sentences. We also perform a causal intervention experiment, investigating whether moving the representation of a premise along these *belief directions* influences the position of an entailed or contradicted sentence along that same direction. We find that the probes we test are generally context sensitive, but that contexts which should not affect the truth often still impact the probe outputs. Our experiments show that the type of errors depend on the layer, the model, and the kind of data. Finally, our results suggest that belief directions are (one of the) causal mediators in the inference process that incorporates in-context information.

## 1 INTRODUCTION

As Large Language Models (LLMs) enjoy increasing mainstream adoption, it becomes more important to understand why they fail in some cases, while excelling in others. Hallucination is a type of failure where the LLM produces grammatical but inaccurate text. Recent work shows that LLM latent spaces contain directions that are predictive of the truth of sentences (Burns et al., 2023; Marks & Tegmark, 2023), and that this enables mitigating hallucination without additional training (Li et al., 2023). The presence of such directions suggests that the model represents sentences as more or less (likely to be) true, resembling a kind of (occurrent) belief. By projecting hidden activations on such *belief directions* we obtain *belief probes*. Such probes accurately identify if a model represents a sentence as true, even in misleading contexts where prompting fails Burns et al. (2023).

Research into belief directions has already shown how they can be used to mitigate factual errors Li et al. (2023), a type of hallucination that can occur independently of context. Another type of hallucination is characterized by inconsistency (Huang et al., 2023). Working towards the mitigation of this type of hallucination requires understanding the impact of context on belief probes. Our experiments investigate the behaviour of *belief probes* on sentences that appear in contexts with other related sentences. This enables us to determine how inferential contexts influence an LLM's assessment of truth-values.

LLMs perform well on tasks which are commonly held to require reasoning (Suzgun et al., 2022). Thus, we expect LLMs' degrees of belief to be sensitive to relevant information provided in context. For example, given a premise $Q$ followed by a hypothesis $H$, such as *"December is during the winter for New York"* and *"In New York, days are shortest in December"*. We might expect a model to represent (potentially at different points) different degrees of belief for these sentences, such as: 1) prior beliefs (not paying attention to the context at all); 2) conditional beliefs where the context is assumed to be truthful; or 3) beliefs that incorporate the model's own assessment of the context's truth. Our experiments vary the truth of premises and hypotheses to determine to what extent each of these are happening.

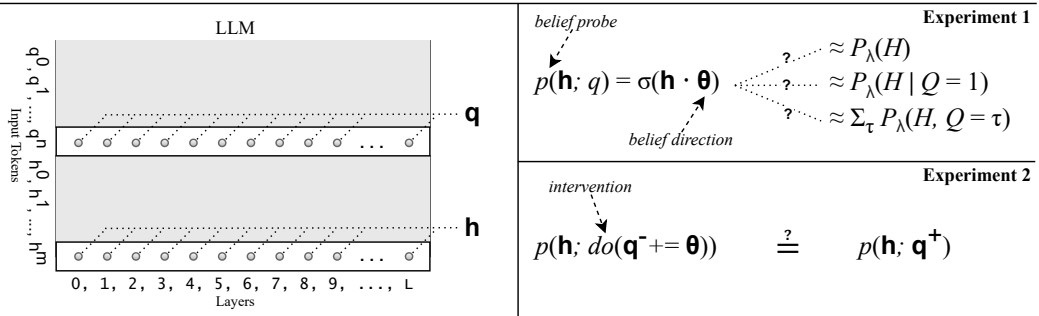

Figure 1: Overview of our setup. LLM representations $\mathbf{q}$ and $\mathbf{h}$ for a premise and hypothesis are extracted and used to train belief probes. In experiment 1, the belief probes are evaluated to determine if and how they incorporate context. In experiment 2, we move a premise's representation in the belief direction, measuring if the probability assigned to the hypothesis changes accordingly.

We also investigate if the belief directions causally mediate truth-value judgment, or if they only reflect the outcome of that process. In other words, we establish whether a representation's positioning along a belief direction determines (in part) where subsequent statements are positioned along the same direction.

We find that belief probes are generally context sensitive, but are also sensitive to irrelevant contexts, and can update probabilities incorrectly. Our results also suggest that the belief directions are (one of the) causal mediators in the inference process that incorporates in-context information.

In summary, our contributions are: (1) experiments demonstrating the context sensitivity of belief probes and the consistency with which they incorporate it; we quantify both across layers, model sizes (7 and 13 billion), type of training (pretrained-only vs. instruction-tuned); and (2) an experiment demonstrating that belief directions causally mediate natural language inference. We also propose a new variant of CCS (Burns et al., 2023) for which convergence is more stable, and otherwise behaves and performs similarly. All our code is available at `https://anonymous.4open.science/r/lcb2-AF5D`.

## 2 RELATED WORK

Probing LLM representations for the truth of sentences has recently received much interest. Burns et al. (2023) introduce Contrast Consistent Search (CCS), an unsupervised probing methods based on the representations of contrasting sentence pairs. Their probes often outperform a (zero-shot) prompting approach, even when applied to misleading prompts. Li et al. (2023) shift model activations in the 'truth direction' at inference time, mitigating hallucination. Their interventions use 1) directions from probes trained with logistic regression (LR) and CCS; and 2) a new method (Mass Mean Shift), which finds the direction as the difference between the means of the true and false sentence representations. Marks & Tegmark (2023) use Mass Mean Shift directions and turn them into probes (Mass-Mean Probing, MMP). They show that all probes (based on LR, CCS, and MMP) generalize well between datasets, with MMP performing the best. In their causal intervention experiment, the representations are moved in the identified directions, and MMP is shown to be the best mediator, causing the highest increase in probability of the model calling a false statement true.

Most of the preceding methods used data consisting of single facts. While Burns et al. (2023) did include datasets with context, including various NLI datasets, they did not study the impact of that context on their probes. We specifically study the in-context behaviour of these probes, analysing their consistency, and what this means for the way LLMs incorporate contextual information.

Like Marks & Tegmark (2023), we also investigate the causal implication of directions in LLM latent space. However, rather than investigate what causes the greatest change in token predictions,

we investigate which direction to move a premise in, such that it causes the correct change in the probability of a related hypothesis, as evaluated by the same direction.

Herrmann & Levinstein (2024) have recently formulated four requirements for a representation to count as belief-like. One of those requirements is *coherence*, which requires that belief probes be logically consistent. Our method measures specific kinds of coherence: two error scores measure the extent to which probed beliefs depend on semantically irrelevant factors, and the two others measure if beliefs are consistent with either kind of context

Recent work has also criticised this type of probing. Belief probes might identify sentence properties that correlate with truth in the model's training data (Levinstein & Herrmann, 2024), especially when truth is not the most salient feature (Farquhar et al., 2023). We argue that arbitrary spurious correlations are unlikely to be coherent, and will perform poorly with our error scores. However, another concern might be that the direction found does encode beliefs, but a more subjective one. Our prompt design does ensure that beliefs are not those held by a person or character that the input text explicitly mentions, as in Farquhar et al. (2023); Zhu et al. (2024). But, future work will have to determine if probed beliefs are best ascribed to: (i) the LLM itself; (ii) a text-producing process simulated by the LLM; (iii) the general public ("commonly-held belief", Levinstein & Herrmann, 2024); or something else entirely. We believe that our method is useful in any case.

## 3 METHODOLOGY

We describe our method in three parts: in 3.1 we cover belief probes, the necessary assumptions, notation, and methods to construct them; in 3.2 we describe how to construct samples suitable to probe for truth-value judgment; and in 3.3 we introduce error scores based on which we evaluate to what extent proper truth-value judgment is taking place.

### 3.1 BELIEF PROBING

We use several belief probing methods in our experiments. These methods use datasets of sentences, consisting of both true and false statements. We can turn any true statement into a false statement (and vice versa) by negating it. We use $X^+$ and $X^-$ to denote the affirmed (positive) and negated (negative) case as $X^-$, and their LLM vector representations are given as $\mathbf{x}^+$, $\mathbf{x}^-$ (see section 4 for how we negate sentences and for how vector representations are extracted). Thus, the dataset used to train probes consist of pairs of hidden states extracted for the positive and negative variants of statements $(\mathbf{x}^+, \mathbf{x}^-, y^+, y^-) \in \mathcal{D}$, and their labels indicating which of the two is true (with $y^+ = 1 - y^-$). When we refer to $X$ or $\mathbf{x}$ without polarity, the polarity could be positive or negative.

When using belief probes, we assume that the truth of sentences is latently modelled by LLMs. We characterize this latent model as a probability distribution $P_\lambda(X)$.[1] The belief probes $p(\mathbf{x})$ are assumed to (approximately) recover this distribution. We use $P_\lambda(x)$ as a shorthand for $P_\lambda(X = 1)$.

We consider only belief probes of the form: $p(\mathbf{x}) = \sigma(\mathbf{x} \cdot \boldsymbol{\theta})$, where $\boldsymbol{\theta}$ is the belief direction.

**Contrast Consistent Search (CCS)** is an unsupervised[2] method with the following objective:

$$\boldsymbol{\theta}_{\text{ccs}} = \arg\min_{\boldsymbol{\theta}} \mathbb{E}_{\mathbf{x}^+, \mathbf{x}^-} \left[ [1 - p(\mathbf{x}^+) - p(\mathbf{x}^-)]^2 + \min\{p(\mathbf{x}^+), p(\mathbf{x}^-)\}^2 \right], \quad (1)$$

which has two terms: the consistency-loss (encouraging solutions where the probabilities add up to one), and the confidence-loss (encouraging non-degenerate solutions, i.e. $p(\mathbf{x}^+) \neq p(\mathbf{x}^-) \neq 0.5$). The objective can be understood as finding a hyperplane with normal $\boldsymbol{\theta}$ that, for each pair: (1) separates $\mathbf{x}^+$ from $\mathbf{x}^-$, and (2) is equidistant to $\mathbf{x}^+$ and $\mathbf{x}^-$.

**Contrast Consistent Reflection (CCR)** is proposed here as a variant of CCS. Rather than finding a hyperplane from which $\mathbf{x}^+$ and $\mathbf{x}^-$ are equidistant, this method requires $\mathbf{x}^+$ and $\mathbf{x}^-$ to be each other's reflection in the hyperplane. It has the following objective:

$$\boldsymbol{\theta}_{\text{ccr}} = \arg\min_{\boldsymbol{\theta}} \mathbb{E}_{\mathbf{x}^+, \mathbf{x}^-} \left[ ||\mathbf{x}^+ - \mathbf{P}\mathbf{x}^-||_2 \right], \quad (2)$$

---

[1]This distribution is entirely separate from the probabilities assigned to tokens by the LM-head.

[2]By unsupervised we mean that no knowledge of which sentences are true or false is given.

where $\mathbf{P} = \mathbf{I} - 2\boldsymbol{\theta}\boldsymbol{\theta}^\intercal$ is the Householder transformation that performs the reflection.

This objective does not share the degenerate solution of CCS. This is because for $p(\mathbf{x}^+) = p(\mathbf{x}^-) = 0.5$, we need $\boldsymbol{\theta} \cdot \mathbf{x}^+ = \boldsymbol{\theta} \cdot \mathbf{x}^- = 0$, and since $|\boldsymbol{\theta}| = 1$ this would imply that $\boldsymbol{\theta}$ is orthogonal to $\mathbf{x}^+$ and $\mathbf{x}^-$. Thus, while they are equidistant in that scenario (a distance of zero), assuming that $\mathbf{x}^+ \neq \mathbf{x}^-$, they will not be each other's reflection.

In our experience CCS does not consistently converge to a good minimum. Burns et al. (2023) train 10 probes and use the probe with the lowest training loss. We find that this procedure nonetheless produces belief directions that vary considerably from layer to layer (for example, see Figure 3b), making it harder to analyse. CCR's objective has one term, and we have found it to achieve similar performance with more stable convergence, without the need to train multiple probes.

**Mass Mean Probing (MMP)**  is a supervised method, which defines the belief direction as the difference between the average of the correct and incorrect statements:

$$\boldsymbol{\theta}_{\mathrm{mm}} = \mathbb{E}_{\mathbf{x},y}[\mathbf{x} \,|\, y = 1] - \mathbb{E}_{\mathbf{x},y}[\mathbf{x} \,|\, y = 0], \tag{3}$$

where $y$ is the truth-value (label) for the statement $X$. We do not include the version of MMP that requires an i.i.d. assumption, because we also evaluate on data the probes were not trained on.

**Logistic Regression (LR)**  is also used to train a supervised probe. The inputs on which we train the LR probes are $\mathbf{x}' = \mathbf{x}^- - \mathbf{x}^+$, i.e. the difference between the negative and positive statements. We use LR without a bias/intercept term:

$$\boldsymbol{\theta}_{\mathrm{lr}} = \arg\min_{\boldsymbol{\theta}} -\mathbb{E}_{\mathbf{x}',y^+} \left[ y^+ \ln \sigma(\boldsymbol{\theta} \cdot \mathbf{x}') + (1 - y^+) \ln (1 - \sigma(\boldsymbol{\theta} \cdot \mathbf{x}')) \right], \tag{4}$$

where $y^+$ is the label for the positive variant of the sample, i.e. whether $X^+$ is true.

## 3.2 PROBING FOR TRUTH-VALUE JUDGMENT

Truth-value judgment (TVJ) tasks are used in language acquisition research to assess children's linguistic competencies. Subjects are "asked to make a bipolar judgment about whether a statement accurately describes a particular situation alluded to in some context or preamble" (Gordon, 1996). TVJ tasks assume the subject has "some conception of the notion of truth in the sense of a correspondence between what is said and the situation referred to" (Gordon, 1996). Using this assumption, the subjects are then asked questions to probe their understanding of various grammatical constructions.

We use TVJ tasks to explore if LLMs have similar notions of truth, specifically, if LLMs represent the truth of sentences in a way that is sensitive to context. The task could be posed the same way it would be posed to a child, asking questions and making inferences about the LLM's competencies based on its answers. However, by using belief probes, we can infer its "answer" directly from the way it represents the input and learn how it changes throughout its layers. To do this, we have a setup as displayed in Figure 1, where the context or preamble consists of a premise $Q$ and the a hypothesis $H$, which serves the role of the 'question'. In this setup we probe the LLM to see if it represents $H$ as true or as false (instead of asking a question). The bracketed parts in $Q$ and $H$ are omitted or included to produce affirmed and negated variants of the sentences ($Q^+, Q^-, H^+, H^-$).

The setup is similar to a natural language inference task. However, we do not directly evaluate a model on its ability to classify sentence pairs by their *meaning relation*: $R \in \{e, c, n\}$ (entailment, contradiction or neutral). Instead, we measure if the model's truth-value judgments (as measured by the belief probes) are consistent with it being able to differentiate between the meaning relations.

In the introduction, we mentioned different ways in which beliefs can interact with the context of a statement. We define three kinds of beliefs in the following way:

- *prior beliefs*, independent of the context, given by $P_\lambda(H)$;
- *conditional beliefs*, specifically where the context is assumed to be truthful, given by $P_\lambda(H|q)$;
- *marginal beliefs*, where the truth of the premise and hypothesis are modeled jointly, with the effect of the premise summed out, given by $\sum_\tau P_\lambda(H, Q{=}\tau)$.[3]

---

[3]We leave this expression unsimplified to distinguish marginal beliefs from prior beliefs and to emphasize the dependence on the joint distribution.

Table 1: Rules for conditional belief probes, and corresponding error scores. The subscript $e$ and $c$ indicate hypotheses entailed or contradicted by their premise.

| | (in)equality | error score |
|---|---|---|
| E1 | $P_\lambda(h\|\tilde{Q}) = P_\lambda(h)$ | $\|p(\mathbf{h};\tilde{q}) - p(\mathbf{h})\| \cdot \|PE^{-1}\|$ |
| E2 | $P_\lambda(h\|Q') \approx P_\lambda(h)$ | $\|p(\mathbf{h};q') - p(\mathbf{h})\| \cdot \|PE^{-1}\|$ |
| E3 | $P_\lambda(h_e\|q^-) \leq P_\lambda(h)$ $P_\lambda(h_c\|q^-) \geq P_\lambda(h)$ | $\max\{(p(\mathbf{h};q^-) - p(\mathbf{h})) \cdot PE^{-1},\ 0\}$ |
| E4 | $\sum_\tau P_\lambda(h, Q^-{=}\tau) \approx \sum_\tau P_\lambda(h, Q^+{=}\tau)$ | $\|p(\mathbf{h};q^-) - p(\mathbf{h};q^+)\| \cdot \|PE^{-1}\|$ |

We can also imagine beliefs in between conditional and marginal, which we can think of as also assigning a probability to the context's truthfulness. These beliefs are candidates for what is measured by a probe $p(\mathbf{h};q)$, i.e. a probe applied to the LLM representation of a hypothesis $H$ when preceded by a premise $Q$. Conditional beliefs, marginal beliefs, and beliefs in between the two, can all be said to embody truth-value judgment, because they are valid ways of incorporating the context.

## 3.3 EVALUATION

To evaluate LLMs on their ability to do truth-value judgment, we include a number of error scores, each indicating the extent to which probe outputs indicate a violation of some desirable behaviour. Table 1 shows the error scores and the (in)equalities on which they are based.

We first define the *premise effect* ($PE$) as the difference in probability assigned to the hypothesis when preceded with an affirmed premise and probability assigned to the hypothesis on its own: $PE = p(\mathbf{h};q^+) - p(\mathbf{h})$. We call the mean absolute premise effect that a method obtains when evaluated its *premise sensitivity*. This metric can help us differentiate between prior beliefs on the one hand, and conditional or marginal beliefs on the other.

The effect of adding the in-context premise can differ in magnitude, depending on which belief probing method we use. In order to make the error scores of different methods comparable to each other, we express the magnitude of the errors in multiples of the premise effect $PE$. This makes the error scores independent of the overall premise sensitivity of the belief probing method.

The first two error scores, E1 and E2 (see Table 1) are based on the fact that we expect the probabilities to only depend on factors actually capable of influencing the truth value of the hypothesis. Thus, these error scores are proportional to the absolute change in probability that occurs after having the hypothesis preceded by either: 1) a corrupted premise $\tilde{q}$, or 2) an unrelated premise $q'$. The truth value of both corrupted and unrelated premises are independent of the truth value of the hypothesis, which is why we should expect the equalities for E1 and E2 in Table 1 to hold.

E3 and E4 measure when probes fail to behave like *conditional* and *marginal* beliefs, respectively.

For E3, we assume the model treats the context as truthful, and thus should consider the premise false when negated (and true when affirmed). If the premise is false, the original meaning relation either switches (between entailment and contradiction), or becomes neutral. When the relationship switches the premise effect should be opposite as well (from increasing the probability, to decreasing, and vice versa). But, if negation creates a neutral relationship, then the probability should be the same as when there is no premise. Together, this gives us the inclusive inequalities in the left column of Table 1. For the error score, we have: $(p(\mathbf{h};q^-) - p(\mathbf{h})) \cdot PE^{-1} = \frac{p(\mathbf{h};q^-)-p(\mathbf{h})}{p(\mathbf{h};q^+)-p(\mathbf{h})}$. By taking the $max\{\cdot, 0\}$ of this fraction, we can isolate those cases where the numerator and denominator have the same sign, which are the errors we want to capture in the score.

For E4, if the language model bases itself on its own evaluation of the premise, then it should ignore whether the premise is affirmed or negated. In that case, the probability assigned to the hypothesis should be equal regardless of the polarity of the premise assertion.

Because E3 and E4 measure deviations for two different types of beliefs, they are opposing and it is impossible to have a score of zero for both simultaneously. See Appendix A for additional details.

## 4 EXPERIMENTS

To answer our research questions, we make use of datasets with samples of related sentences, whose truth values depend on each other. We use samples from these datasets by creating prompts where the sentences are either affirmed or negated.

### TRAINING & EVALUATING PROBES

We train probes in two settings: `no-prem` and `pos-prem`. For `no-prem`, the premise $Q$ is left out, and for `pos-prem` the premise appears in the positive (or affirmed) variant. We include these settings, because they allow us to understand more about how beliefs are represented. A belief direction found in the `no-prem` setting, that separates true from false (on held out data from the same distribution), is a direction that represents prior belief. If that direction also shows context-sensitivity (when evaluated with premises in-context), that would be evidence that the model does not represent the prior and contextual beliefs in orthogonal directions. For `pos-prem`, the direction(s) that separate true from false in the training distribution can also be influenced by what appears in context. If the directions found for `pos-prem` and `no-prem` are different, it suggests there *are* separate (but possibly related) directions used to represent context-sensitive truth.

The probe inputs $\mathbf{h}$ are the mean-normalized representations of the answer tokens ('correct'/'incorrect') of the sample, extracted for each layer. To make the results from different probing methods comparable, we calibrate the probes such that their predictions for the $p(\mathbf{h})$ case have the same variance. We train probes on the following LLMs: Llama2-7b, Llama2-13b (Touvron et al., 2023), and OLMo-7b with and without instruction tuning (Groeneveld et al., 2024).

To measure the premise effect, and error scores described in subsection 3.3, we include the following evaluation cases: $p(\mathbf{h})$, $p(\mathbf{h}; q^+)$, $p(\mathbf{h}; q^-)$, $p(\mathbf{h}; q')$, $p(\mathbf{h}; \tilde{q})$. We evaluate both the `no-prem` and `pos-prem` in all of these cases. The first two cases are 'in distribution' for the `no-prem` and `pos-prem` settings, respectively. The other combinations are out of distribution. When evaluating the probes we use: $p(\mathbf{h}) = \frac{1}{2}(1 - p(\mathbf{h}^-) + p(\mathbf{h}^+))$.

### DATA

We use two existing datasets in our experiments. The first dataset (SNLI, Bowman et al., 2015) contains statements that describe images (to which an LLM has no access). The second dataset (EntailmentBank, Dalvi et al., 2021) contains hypotheses that are sentences with general world knowledge. These are facts the LLM may have encountered during training and for which it could already have a strong prior belief.

For both datasets, the polarity of the premises and hypothesis is determined by the inclusion or omission of the 'in' that appears in square brackets. This style of negation sidesteps potential problems with choosing how to negate a sentence, which can sometimes be difficult.[4] The corrupted sentences are created by replacing the characters in each word of the base sentence with random characters.

**EntailmentBank** This dataset is similar in structure to SNLI, consisting of premises and hypotheses, but it contains only entailments. The subject of the statements are also different since EntailmentBank was derived from ARC (Clark et al., 2018), which consists of grade-school level science questions. We combine the premises of EntailmentBank with the questions and answers from ARC on which they were based. In order to create contradictions we combine the premises of EntailmentBank with an incorrectly answered question. For example:

> You are given the following question:
> > In New York, the shortest period of daylight occurs during? (A) December (B) June

$Q_a$    The statement "New York is located in the northern hemisphere." is [in]correct.
$Q_b$    The statement "December is during the winter for New York." is [in]correct.
$H$    Answering the question with "(B) June" is [in]correct.

The answer "June" is incorrect, and thus $H$ contradicts the information in $Q_a, Q_b$ (when it is not

---

[4]For example, negating "four children are playing in some water" as "four children are not playing in some water", still presupposes the existence of four children. Using a negative meta statement leaves open the possibility that the presupposition is false (e.g. the number of children is inaccurate).

negated), while in the sample with the correct answer $H$ would be entailed by $Q_a, Q_b$. The dataset contains trees of entailing sentences, where each premise may itself be supported by premises of its own. However, we disregard anything but the first level of supporting premises. For the $p(\mathbf{h}; q)$ case, we use the distractor premises provided in the dataset. These were ranked (Dalvi et al., 2021) as potentially relevant, but during annotation were not selected to be part of the entailment tree.

**SNLI** This dataset is a Natural Language Inference dataset, it consists of premise-hypothesis pairs, which are labeled as: entailment, contradiction, or neutral. These labels describe the meaning relation $R$ between the sentences. The samples for this dataset were created based on the descriptions of images. To avoid ambiguity, we establish a context as follows:

> You are looking at a picture (A) which is placed next to an unrelated picture (B).
> $Q$    Describing picture {A/B} as: "Four children are playing in some water." is [in]correct.
> $H$    Saying (about picture A) that: "The children are wet." is [in]correct.

The neutral sentences for the $p(\mathbf{h}; q')$ case are obtained by taking the premise from a different, randomly sampled premise-hypothesis pair. Furthermore, for this case, the 'A/B' that appears in curly brackets is set to B to ensure that there is a fully neutral relationship. Without it, the fact that the two sentences are about the same picture could make their (simultaneous) truth less likely. It is also set to B for $p(\mathbf{h}; \tilde{q})$, and set to A for all other cases.

Because the model does not have access to the picture, its prior belief should result in 50% accuracy. However, for SNLI it is possible to predict the label solely from the hypothesis Poliak et al. (2018). This makes for an interesting scenario when it comes to belief probing. A belief probing method might identify a direction that only encapsulates a statistical pattern, rather than the model's belief direction. Although, it is also possible that the statistical pattern is represented the same way as other reasons to believe a sentence, in the model's belief direction. After the addition of a premise, we do not expect a representation should move (coherently) in a direction which merely encodes a statistical pattern. Thus, if a probe trained only on hypotheses *does* respond coherently to the presence of a premise at test time, it suggests that we have found a belief direction.

### 4.1 Effect of altering premises

We evaluate the probes on held-out data, including data from all the other variants. We also include an additional baseline, based on the model's LM-head, where the probabilities assigned to the 'correct'/'incorrect' tokens are rescaled to sum up to one.

#### Results

Table 2 gives an overview of the average probabilities for $p(\mathbf{h}; q^+)$, $p(\mathbf{h}; q^-)$, and $p(\mathbf{h})$, split by whether the premise-hypothesis pair had an entailment or contradiction relation. We observe that the probabilities assigned to hypotheses depend strongly on the presence of relevant premises. When the hypothesis is entailed, the probabilities are higher, when the hypothesis is contradicted they are lower. This is even true for probes trained without the premises present (`no-prem`), although the sensitivity to premises is lower. Most `no-prem` probes also achieve good accuracy for $p(\mathbf{h}; q^+)$, showing that the direction encodes more than just prior beliefs.

**Error scores.** In Table 2, we can see that especially E1 and E2 are quite high. This suggests that belief directions are sensitive to irrelevant information. Probes trained on `no-prem` often have E1 and E2 close to one. Because the error scores are normalized by the premise effect, a value of one means that, on average, a corrupted or unrelated premise has an effect with the same magnitude as the original affirmed premise. The error scores improve when probes are trained on `pos-prem`. Comparing Llama2-7b to Llama2-13b (see Table B.2) shows the scores are not consistently lower for the larger model, meaning error scores show no sign of scaling with model size.

**Spurious correlations.** Looking at SNLI, both LR and MMP show premise sensitivity, suggesting that they find directions indicative of more than just the spurious correlations present in the hypotheses of SNLI. However, for LR the probe's behaviour does seem affected by the spurious correlations. Its average probabilities for samples with negated premises is not between the probabilities obtained for samples with positive premises and no premises, resulting in a high E3+E4 score.

Table 2: Accuracy of $p(\mathbf{h};q^+)$ (Acc), mean probabilities (orange=0, gray=0.5, blue=1), and trimmed mean errors scores for probes of each method on both datasets for Llama2-7b. The probes are from layers (L) with: (1) the best accuracy; and (2) the overall lowest error scores (by average error rank $E*$). The best scores per dataset are in bold, for E3 and E4 the bold values are based on their sum. CCS omitted, full table in Appendix B.

| | | Method | L | Acc | $E*$ | Entailment $p(\mathbf{h};q^+)$ | $p(\mathbf{h};q^-)$ | $p(\mathbf{h})$ | Contradiction $p(\mathbf{h};q^-)$ | $p(\mathbf{h};q^+)$ | E1 | E2 | E3 | E4 |
|---|---|---|---|---|---|---|---|---|---|---|---|---|---|---|
| EntailmentBank | no-prem | LM-head | - | .80 | 145.8 | .61 | .52 | .50 | .49 | .38 | .96 | .90 | .31 | 1.11 |
| | | CCR | 14 | .63 | 141.4 | .55 | .52 | .49 | .48 | .45 | 1.04 | 1.22 | .99 | .62 |
| | | | 29 | .58 | 127.4 | .53 | .51 | .49 | .48 | .46 | .93 | 1.17 | .86 | .74 |
| | | LR | 16 | .93 | 160.0 | .78 | .59 | .50 | .41 | .24 | 1.04 | .90 | .21 | 1.36 |
| | | | 14 | .92 | 107.6 | .75 | .61 | .50 | .39 | .25 | .89 | .85 | .28 | 1.15 |
| | | MMP | 19 | .89 | 145.2 | .71 | .54 | .49 | .46 | .31 | .68 | .79 | .20 | 1.28 |
| | | | 22 | .86 | 103.6 | .69 | .53 | .49 | .47 | .33 | .71 | .83 | .31 | 1.17 |
| | pos-prem | CCR | 16 | .87 | 89.0 | .86 | .54 | .50 | .46 | .18 | .56 | .67 | .05 | 1.27 |
| | | | 14 | .86 | 70.0 | .84 | .52 | .50 | .49 | .18 | .57 | .65 | .05 | 1.27 |
| | | LR | 18 | **.96** | 51.6 | .92 | .60 | .50 | .40 | .10 | .52 | .58 | **.08** | **1.16** |
| | | | 14 | .95 | **43.6** | .91 | .60 | .49 | .41 | .11 | **.43** | **.56** | **.08** | **1.16** |
| | | MMP | 14 | .89 | 60.6 | .86 | .52 | .50 | .49 | .16 | .51 | .61 | .04 | 1.26 |
| | | | 14 | .89 | 60.6 | .86 | .52 | .50 | .49 | .16 | .51 | .61 | .04 | 1.26 |
| SNLI | no-prem | LM-head | - | .62 | 150.6 | .57 | .54 | .52 | .43 | .43 | .89 | .88 | .36 | 1.35 |
| | | CCR | 7 | .57 | 138.8 | .52 | .52 | .53 | .49 | .49 | .93 | 1.02 | 1.16 | .26 |
| | | | 12 | .52 | 100.2 | .51 | .53 | .51 | .47 | .50 | .74 | .95 | .99 | .27 |
| | | LR | 13 | .85 | 189.8 | .67 | .75 | .50 | .24 | .32 | .91 | 1.13 | .89 | 1.13 |
| | | | 20 | .75 | 103.4 | .65 | .57 | .50 | .42 | .35 | .72 | .96 | .37 | 1.21 |
| | | MMP | 13 | .88 | 178.2 | .61 | .65 | .50 | .35 | .38 | .91 | 1.06 | 1.03 | .54 |
| | | | 32 | .45 | 129.0 | .48 | .51 | .51 | .49 | .52 | .92 | 1.04 | .68 | .87 |
| | pos-prem | CCR | 26 | .91 | 53.8 | .87 | .68 | .50 | .28 | .14 | .42 | .53 | .47 | .60 |
| | | | 28 | .91 | 53.6 | .86 | .70 | .50 | .28 | .14 | .41 | .51 | .49 | .57 |
| | | LR | 16 | **.95** | 95.6 | .93 | .77 | .51 | .22 | .06 | .47 | .61 | .63 | .42 |
| | | | 26 | .95 | **41.8** | .88 | .68 | .50 | .29 | .11 | **.38** | **.48** | .44 | .61 |
| | | MMP | 17 | .94 | 90.0 | .92 | .77 | .50 | .20 | .09 | .46 | .57 | **.68** | **.35** |
| | | | 6 | .74 | 49.6 | .69 | .65 | .50 | .34 | .27 | .39 | .50 | .62 | .44 |

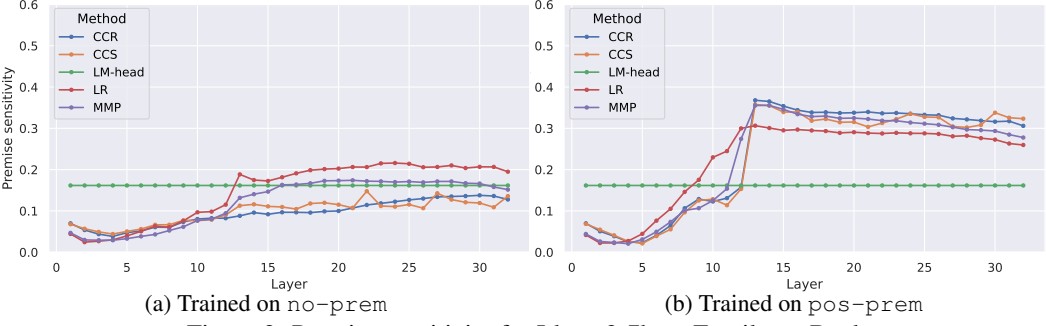

(a) Trained on `no-prem`  (b) Trained on `pos-prem`

Figure 2: Premise sensitivity for Llama2-7b on EntailmentBank.

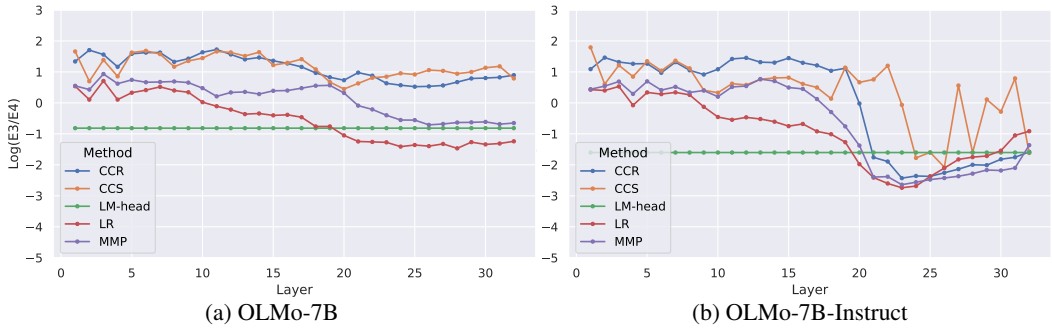

(a) OLMo-7B  (b) OLMo-7B-Instruct

Figure 3: Log-ratio of E3 and E4 error score for probes trained using the `no-prem` variants of EntailmentBank on OLMo-7B and OLMo-7B-Instruct.

**LM-head baseline.** Most probes beat the LM-head both in terms of accuracy and premise sensitivity. This suggests that inconsistency hallucinations can occur even when the LLM's representations contain information able to prevent it. This is in line with findings for non-contextual hallucination.

**Premise sensitivity by layer.** Figure 2 shows the premise sensitivity across layers for probes of each method when applied to Llama2-7b. These were trained on the `no-prem` (left) and `pos-prem` (right) variants of the EntailmentBank data. We again see that all methods show a degree of premise sensitivity in all cases, with `no-prem` showing less premise sensitivity than `pos-prem`. There do not seem to be layers where the probe is not sensitive to the premises (approximating $P_\lambda(H)$), while still having above random accuracy (see subsection C.2). Suggesting that LLMs do not represent prior beliefs $P_\lambda(H)$ fully independently.

**Pretrained-only vs. instruction-tuned.** Figure 3 In the later layers of the instruction-tuned model, it leans more toward E4 errors. This indicates that the instruct-tuned model's behaviour is a lot more sensitive to whether the premise is negated or affirmed. This suggests that instruction-tuning makes the model more likely to represent prior assertions as true, which is in line with the instruction-tuning objective.

## 4.2 INTERVENING ON PREMISE BELIEFS

In this experiment, we alter the LLM's internal representations directly, rather than only altering the input data. We take the belief directions found by probing methods in the first experiment, and move the representations of the premises along this direction.

We perform this experiment for the $p(\mathbf{h}; q^+)$ and $p(\mathbf{h}; q^-)$ cases. We move the premise in the belief direction found during `pos-prem` training, and use that same direction to evaluate pre-intervention: $p(\mathbf{h}; q)$, and post-intervention: $p(\mathbf{h}; do(\mathbf{q}\pm=\boldsymbol{\theta}))$. We perform the intervention

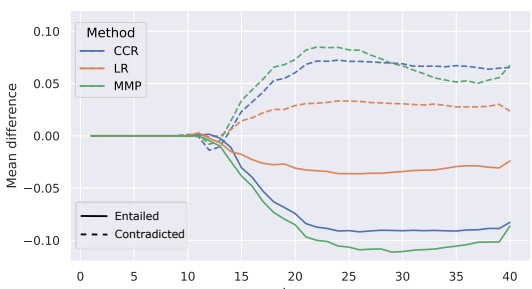

Figure 4: Effect of intervention: mean difference in probability $p(\mathbf{h}; do(\mathbf{q}^+-=\boldsymbol{\theta})) - p(\mathbf{h}; q^+)$ over layers for entailments and contradictions.

using the same method and parameters as Marks & Tegmark (2023). The intervention is done on Llama2-13b in layers 8-14, and applied to the representations of the answer tokens (correct, incorrect), and the period after. All interventions have the same magnitude: $|\boldsymbol{\theta}_{\mathrm{mm}}|$.

**Results.** In Figure 4, we can see the effect of the causal intervention for the $p(\mathbf{h}; q^+)$ case. When we move the affirmed premises backwards in the belief direction, the probabilities of entailed hypotheses decrease and the probabilities of contradicted hypotheses increase, exactly as expected. This shows that belief directions causally mediate the incorporation of in-context information. We see that intervening with the direction found by LR has a smaller effect than MMP and CCR. The largest change is a reduction of around ten percentage points for entailed hypotheses. See Figure C.1 for the results of $p(\mathbf{h}, do(\mathbf{q}^-+=\boldsymbol{\theta}))$.

## 5 CONCLUSION

We have investigated LLM truth-value judgment, which requires correctly incorporating context when determining the truth value of a sentence. Based on our expectations of how the probability of a sentence should or should not change in a supporting, contradicting, or neutral context, we created four error scores. In our experiments, we used several probing methods on four language models, and quantified how they assign probabilities to hypotheses in different contexts.

From our results it is clear that LLMs do incorporate context when representing sentences as more or less (likely to be) true. However, we also observe that contexts which should have no bearing on truth values still have a sizeable impact on a sentence's position along the belief direction revealed by the probes. Our intervention experiment shows that the positioning of premises along belief directions (partially) determines the positioning of related hypotheses along the same direction.

We believe that our work is a first step to better understanding and addressing inconsistencies in LLM generated text. Fully understanding the in-context behaviour of belief-probing methods will help to ascertain exactly why inconsistent generations arise, for example whether: the model has represented part of the context as false; the model fails to accurately represent the meaning relation between the context and possible generations; or both. Finally, the causal connection between truth values of related sentences might be part of a mechanism that, when fully uncovered, could explain how LLMs do well on reasoning tasks.

### LIMITATIONS AND FUTURE WORK

In our experiments we have investigated one direction at a time. Recently, Bürger et al. (2024) have shown that beliefs in LLMs use a two-dimensional subspace: one direction consistently points from true to false, and another is polarity-sensitive and points from false to true for negated statements. It is possible that marginal and conditional beliefs also occupy independent directions, but finding them requires data where the '*being entailed / contradicted by context*' and '*being true / false*' features can be varied completely independently. We leave this for future work.

We would also like to dive deeper into the representations of meaning-relations in LLMs, and the exact mechanisms responsible for incorporating that information into the belief directions. For example, by investigating the construction of probes that reveal if a model represents two sentences as having a particular meaning relation. Then, we can detect when the model disagrees with the gold standard meaning relation provided by the dataset. With probabilities for all three relevant variables: $H$, $Q$, $R$, an even more precise evaluation would become possible.

In our experiments we have only investigated models with 7 or 13 billion parameters. To fully investigate the interaction of our error scores with model size, additional experiments are needed.

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

# A    ERROR SCORES

Here we try to give some (geometric) intuitions for our error scores. Specifically, we make use of the diagrams presented in Figure A.1. These diagrams take as a baseline the probability assigned to the hypothesis on its own $p(\mathbf{h})$, and show all other probabilities relative to it. The diagram assumes we are looking at premise-hypothesis pairs with entailment relations. The diagrams for contradictions would be identical, but mirrored vertically.

E1 and E2 consistency errors are shown in box A in Figure A.1. Both of these errors involve the difference in probability assigned to (a) the hypothesis on its own and (b) the hypothesis preceded with an irrelevant statement, which is either:

- a premise where the characters have been replaced by random characters $p(\mathbf{h}; \tilde{q})$; or
- a premise that has been replaced by another randomly sampled premise $p(\mathbf{h}; q')$.

See Appendix D for examples.

E3 and E4 consistency errors are indicative of two opposing behaviours potentially exhibited by a language model. E3 assumes that the context (containing the premise) is truthful, and that what is asserted should be taken at face value. If a contradicting premise is (said to be) true this should reduce the probability assigned to the hypothesis, and if a supporting premise is (said to be) true it should increase the probability assigned to the hypothesis. On the other hand, E4 is assumes that the model uses its own evaluation of the context, ignoring if it is asserted to be true or false. If this is the case, then the probability assigned to the hypothesis should not depend on the truth value that is asserted of the premise. These two are displayed in three different scenarios (B, C, D) in Figure A.1.

In B, we have $p(\mathbf{h}) < p(\mathbf{h}; q^-) < p(\mathbf{h}; q^+)$, in this scenario it is always the case that $E3 + E4 = 1$ (recall that the error scores are given as multiples of $PE = p(\mathbf{h}; q^+) - p(\mathbf{h})$). When evaluating the overall consistency of the model this is the best score for $E3 + E4$ that we can expect.

In C, we have $p(\mathbf{h}) < p(\mathbf{h}; q^+) < p(\mathbf{h}; q^-)$, this scenario is 'double wrong', in that there is now a part of the probability that is punished by both error scores. Regardless of whether the model trusts that the context is truthful or trusts itself, it should never give a higher probability to an entailed hypothesis after seeing the premise negated than when it saw it affirmed.

In D, we have $p(\mathbf{h}; q^-) < p(\mathbf{h}) < p(\mathbf{h}; q^+)$, now we have E3 equal to zero, since it is perfectly acceptable

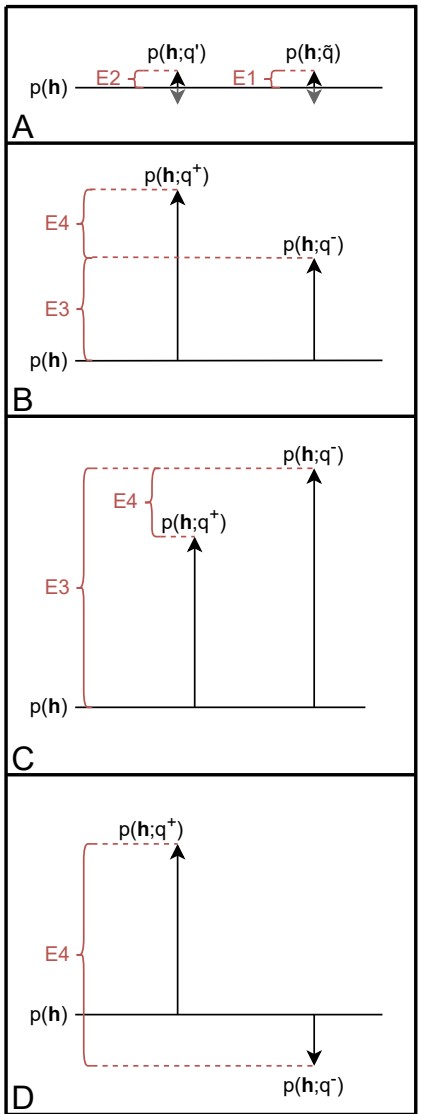

Figure A.1: Error score diagram.

for the probability of the hypothesis to decrease when preceded by a negated supporting premise. This can occur in two ways, either the supporting premise became a contradicting premise and thus makes the hypothesis less likely, or the premise became neutral, in which case it still takes away one (potentially important) reason to believe the hypothesis.

## B  ADDITIONAL TABLES

### B.1  LLAMA2-7B

| | | Method | L | Acc | E* | Entailment | | | Contradiction | | E1 | E2 | E3 | E4 |
|---|---|---|---|---|---|---|---|---|---|---|---|---|---|---|
| | | | | | | $p(\mathbf{h};q^+)$ | $p(\mathbf{h};q^-)$ | $p(\mathbf{h})$ | $p(\mathbf{h};q^-)$ | $p(\mathbf{h};q^+)$ | | | | |
| EntailmentBank | | LM-head | - | .80 | 214.0 | .61 | .52 | .50 | .49 | .38 | .96 | 0.90 | .31 | 1.11 |
| | no-prem | CCR | 14 | .63 | 141.4 | .55 | .52 | .49 | .48 | .45 | 1.04 | 1.22 | .99 | .62 |
| | | | 29 | .58 | 127.4 | .53 | .51 | .49 | .48 | .46 | .93 | 1.17 | .86 | .74 |
| | | CCS | 19 | .71 | 241.0 | .58 | .52 | .50 | .48 | .42 | .95 | 1.08 | .79 | .91 |
| | | | 22 | .34 | 170.6 | .45 | .49 | .50 | .50 | .55 | .87 | .97 | .89 | .50 |
| | | LR | 16 | .93 | 160.0 | .78 | .59 | .50 | .41 | .24 | 1.04 | .90 | .21 | 1.36 |
| | | | 14 | .92 | 107.6 | .75 | .61 | .50 | .39 | .25 | .89 | .85 | .28 | 1.15 |
| | | MMP | 19 | .89 | 145.2 | .71 | .54 | .49 | .46 | .31 | .68 | .79 | .20 | 1.28 |
| | | | 22 | .86 | 103.6 | .69 | .53 | .49 | .47 | .33 | .71 | .83 | .31 | 1.17 |
| | pos-prem | CCR | 16 | .87 | 89.0 | .86 | .54 | .50 | .46 | .18 | .56 | .67 | .05 | 1.27 |
| | | | 14 | .86 | 70.0 | .84 | .52 | .50 | .49 | .18 | .57 | .65 | .05 | 1.27 |
| | | CCS | 28 | .91 | 121.4 | .86 | .56 | .50 | .44 | .15 | .48 | .55 | .05 | 1.20 |
| | | | 14 | .89 | 83.0 | .87 | .54 | .50 | .46 | .15 | .54 | .63 | .06 | 1.21 |
| | | LR | 18 | **.96** | 51.6 | .92 | .60 | .50 | .40 | .10 | .52 | .58 | **.08** | **1.16** |
| | | | 14 | .95 | **43.6** | .91 | .60 | .49 | .41 | .11 | **.43** | **.56** | **.08** | **1.16** |
| | | MMP | 14 | .89 | 60.6 | .86 | .52 | .50 | .49 | .16 | .51 | .61 | .04 | 1.26 |
| | | | 14 | .89 | 60.6 | .86 | .52 | .50 | .49 | .16 | .51 | .61 | .04 | 1.26 |
| SNLI | | LM-head | - | .62 | 150.6 | .57 | .54 | .52 | .43 | .43 | .89 | .88 | .36 | 1.35 |
| | no-prem | CCR | 7 | .57 | 138.8 | .52 | .52 | .53 | .49 | .49 | .93 | 1.02 | 1.16 | .26 |
| | | | 12 | .52 | 100.2 | .51 | .53 | .51 | .47 | .50 | .74 | .95 | .99 | .27 |
| | | CCS | 12 | .73 | 164.8 | .55 | .53 | .48 | .47 | .45 | .83 | .92 | .96 | .36 |
| | | | 18 | .34 | 162.2 | .48 | .49 | .51 | .51 | .52 | .78 | .91 | .96 | .22 |
| | | LR | 13 | .85 | 189.8 | .67 | .75 | .50 | .24 | .32 | .91 | 1.13 | .89 | 1.13 |
| | | | 20 | .75 | 103.4 | .65 | .57 | .50 | .42 | .35 | .72 | .96 | .37 | 1.21 |
| | | MMP | 13 | .88 | 178.2 | .61 | .65 | .50 | .35 | .38 | .91 | 1.06 | 1.03 | .54 |
| | | | 32 | .45 | 129.0 | .48 | .51 | .51 | .49 | .52 | .92 | 1.04 | .68 | .87 |
| | pos-prem | CCR | 26 | .91 | 53.8 | .87 | .68 | .50 | .28 | .14 | .42 | .53 | .47 | .60 |
| | | | 28 | .91 | 53.6 | .86 | .70 | .50 | .28 | .14 | .41 | .51 | .49 | .57 |
| | | CCS | 13 | **.95** | 159.2 | .97 | .79 | .50 | .23 | .08 | .52 | .65 | **.66** | **.36** |
| | | | 26 | .88 | 65.4 | .85 | .74 | .51 | .25 | .15 | **.38** | .50 | .62 | .43 |
| | | LR | 16 | **.95** | 95.6 | .93 | .77 | .51 | .22 | .06 | .47 | .61 | .63 | .42 |
| | | | 26 | .95 | **41.8** | .88 | .68 | .50 | .29 | .11 | **.38** | **.48** | .44 | .61 |
| | | MMP | 17 | .94 | 90.0 | .92 | .77 | .50 | .20 | .09 | .46 | .57 | .68 | .35 |
| | | | 6 | .74 | 49.6 | .69 | .65 | .50 | .34 | .27 | .39 | .50 | .62 | .44 |

Table B.1: Accuracy (Acc), mean probabilities (orange=0, gray=0.5, blue=1), and errors scores for probes of each method on both datasets. The probes are from layers (L) with: (1) the best probe accuracy; and (2) the overall lowest error scores (by average error rank $E*$).

| | Method | L | Acc | $E*$ | Entailment | | | Contradiction | | E1 | E2 | E3 | E4 |
|---|---|---|---|---|---|---|---|---|---|---|---|---|---|
| | | | | | $p(\mathbf{h};q^+)$ | $p(\mathbf{h};q^-)$ | $p(\mathbf{h})$ | $p(\mathbf{h};q^-)$ | $p(\mathbf{h};q^+)$ | | | | |
| **EntailmentBank** | LM-head | - | .88 | 233.8 | .61 | .58 | .49 | .42 | .37 | 1.38 | 1.18 | .60 | 1.50 |
| no-prem | CCR | 21 | .94 | 232.0 | .71 | .55 | .50 | .45 | .31 | 1.67 | 1.38 | .69 | 1.42 |
| | | 9 | .58 | 135.8 | .52 | .52 | .49 | .47 | .47 | 1.01 | 1.16 | .95 | .25 |
| | LR | 17 | .93 | 250.8 | .70 | .61 | .50 | .40 | .31 | 1.80 | 1.45 | .63 | 1.34 |
| | | 9 | .63 | 125.0 | .56 | .57 | .49 | .40 | .42 | 1.04 | 1.06 | .66 | .84 |
| | MMP | 20 | .94 | 207.4 | .72 | .57 | .50 | .43 | .30 | 1.48 | 1.20 | .49 | 1.39 |
| | | 9 | .63 | 123.4 | .55 | .55 | .48 | .43 | .43 | .93 | 1.11 | .83 | .41 |
| pos-prem | CCR | 19 | .92 | 98.4 | .85 | .59 | .50 | .41 | .19 | .79 | .66 | .08 | 1.35 |
| | | 15 | .90 | 60.2 | .84 | .59 | .50 | .41 | .17 | .65 | .61 | .08 | 1.27 |
| | LR | 17 | .98 | 63.8 | .90 | .67 | .50 | .34 | .12 | .54 | .48 | .13 | 1.00 |
| | | 15 | .97 | 36.4 | .90 | .66 | .51 | .35 | .12 | .56 | .51 | .12 | 1.02 |
| | MMP | 17 | .93 | 98.2 | .86 | .58 | .50 | .42 | .17 | .70 | .60 | .07 | 1.33 |
| | | 15 | .92 | 56.6 | .85 | .59 | .50 | .41 | .16 | .64 | .59 | .08 | 1.24 |
| **SNLI** | LM-head | - | .87 | 247.0 | .59 | .61 | .49 | .36 | .35 | 1.25 | 1.10 | .83 | .85 |
| no-prem | CCR | 21 | .82 | 163.6 | .58 | .54 | .49 | .46 | .41 | .87 | 1.03 | .89 | .44 |
| | | 13 | .69 | 154.0 | .53 | .51 | .51 | .49 | .47 | .89 | .97 | 1.00 | .27 |
| | LR | 19 | .87 | 229.4 | .68 | .66 | .50 | .31 | .29 | 1.07 | 1.07 | .70 | 1.02 |
| | | 4 | .58 | 143.8 | .54 | .55 | .50 | .44 | .45 | .78 | 1.04 | .79 | .47 |
| | MMP | 19 | .89 | 189.4 | .64 | .55 | .50 | .43 | .34 | .92 | .97 | .74 | .74 |
| | | 24 | .88 | 140.6 | .65 | .57 | .51 | .42 | .32 | .79 | .89 | .67 | .77 |
| pos-prem | CCR | 15 | .92 | 115.6 | .91 | .69 | .51 | .28 | .10 | .40 | .53 | .49 | .55 |
| | | 8 | .70 | 73.6 | .68 | .63 | .52 | .38 | .33 | .38 | .48 | .47 | .56 |
| | LR | 18 | .98 | 93.0 | .93 | .73 | .51 | .26 | .06 | .39 | .54 | .47 | .57 |
| | | 17 | .98 | 51.6 | .94 | .70 | .51 | .29 | .06 | .38 | .51 | .39 | .67 |
| | MMP | 18 | .94 | 109.4 | .89 | .66 | .51 | .32 | .11 | .50 | .64 | .40 | .70 |
| | | 4 | .69 | 68.2 | .64 | .53 | .50 | .47 | .34 | .40 | .50 | .08 | 1.13 |

Table B.2: Accuracy (Acc), mean probabilities (orange=0, gray=0.5, blue=1), and errors scores for probes of each method on both datasets. The probes are from layers (L) with: (1) the best probe accuracy; and (2) the overall lowest error scores (by average error rank $E*$).

## C ADDITIONAL FIGURES

### C.1 CAUSAL EXPERIMENT MOVING NEGATED PREMISES TOWARD BELIEF DIRECTION

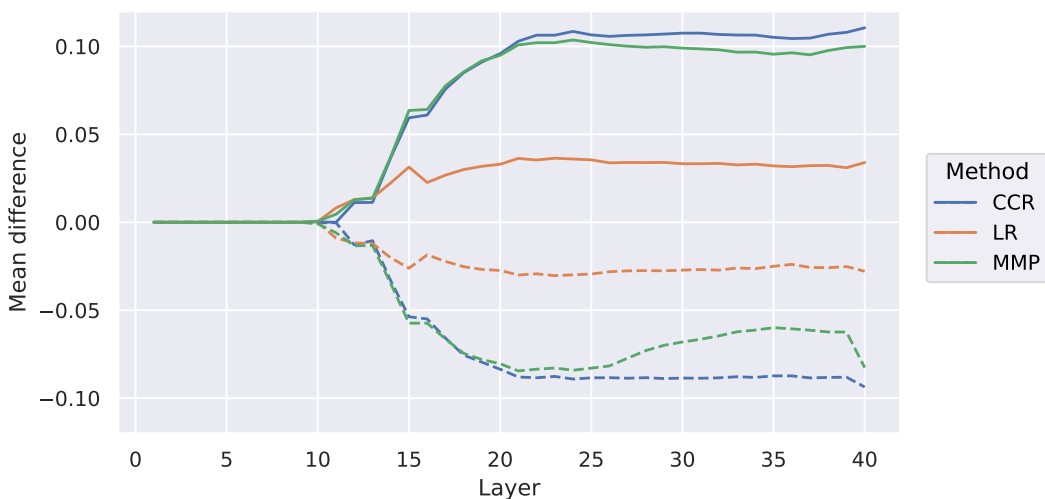

Figure C.1: Mean difference in probability $p(\mathbf{h}; do(\mathbf{q}^- += \boldsymbol{\theta})) - p(\mathbf{h}; q^-)$ after moving negated premises in the positive belief direction.

### C.2 PREMISE SENSITIVITY AND ACCURACY

LLAMA2-7B

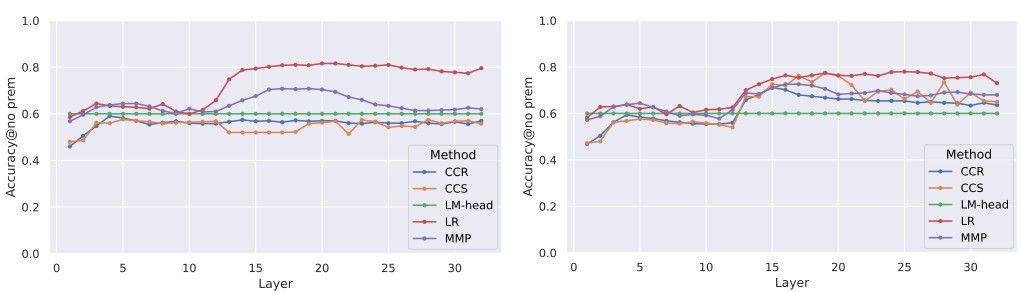

Figure C.2: Llama2-7b - EntailmentBank - Accuracy on `no-prem`. Probes trained on `no-prem` (left) and `pos-prem` (right).

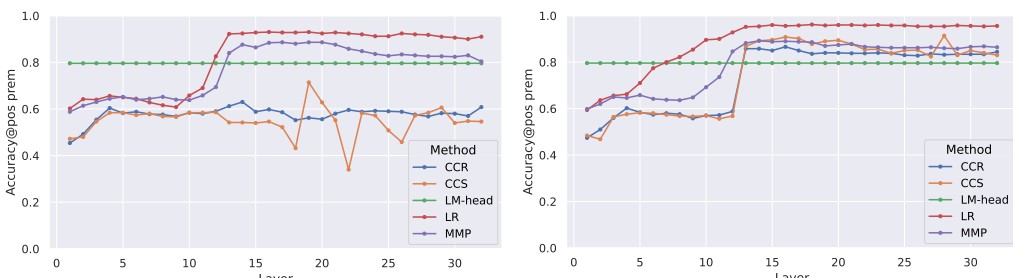

Figure C.3: Llama2-7b - EntailmentBank - Accuracy on `pos-prem`. Probes trained on `no-prem` (left) and `pos-prem` (right).

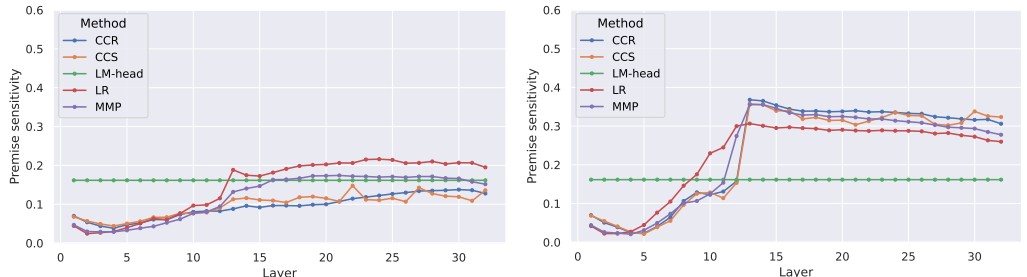

Figure C.4: Llama2-7b - EntailmentBank - Premsise sensitivity. Probes trained on `no-prem` (left) and `pos-prem` (right).

OLMO-7B

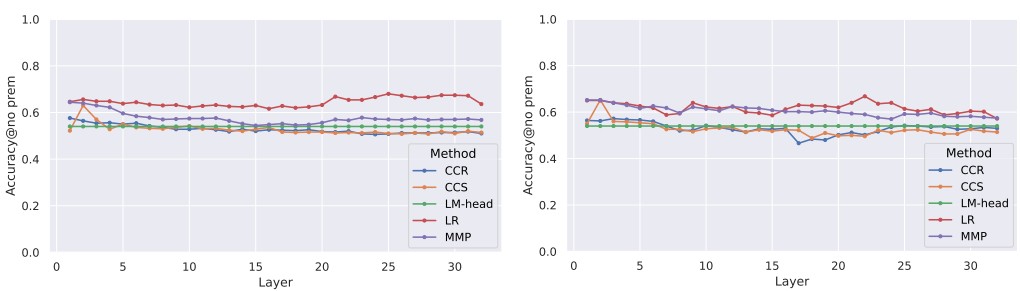

Figure C.5: OLMo-7b - EntailmentBank - Accuracy on `no-prem`. Probes trained on `no-prem` (left) and `pos-prem` (right).

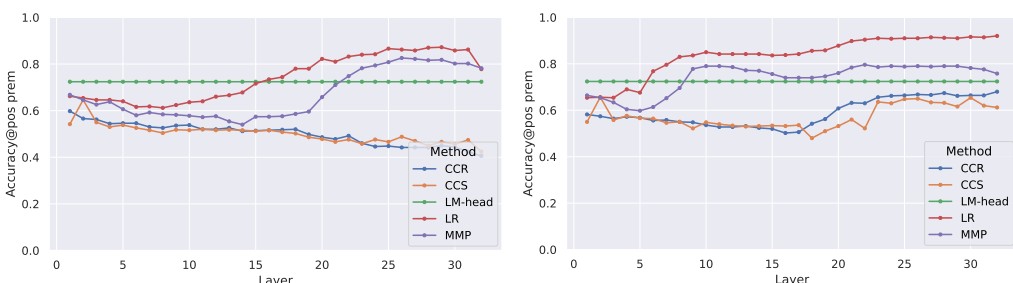

Figure C.6: OLMo-7b - EntailmentBank - Accuracy on `pos-prem`. Probes trained on `no-prem` (left) and `pos-prem` (right).

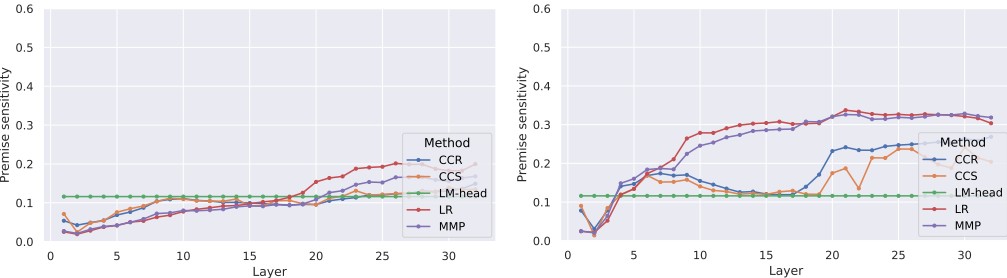

Figure C.7: OLMo-7b - EntailmentBank - Premsise sensitivity. Probes trained on `no-prem` (left) and `pos-prem` (right).

# D  DATA SAMPLES

Each variant of five samples from each dataset. Highlighted in red is the text that is inserted to convert a positive sample $X^+$ into a negative sample $X^-$.

## D.1  ENTAILMENTBANK - `NO-PREM`

1 You are given the following question:
2 > In clear weather, a bright light can be seen for a long distance. In conditions of heavy fog, the visibility is greatly reduced. Which of the following explains the reduced visibility? (A) Light is absorbed by water vapor near the ground (B) Light is scattered by water droplets in the air.
3 Answering the question with "(B) Light is scattered by water droplets in the air." is **in**correct

1 You are given the following question:
2 > The main function of a tree's trunk is to provide (A) air (B) support
3 Answering the question with "(B) support" is **in**correct

1 You are given the following question:
2 > The temperature in a hot star is high enough to pull electrons away from atoms. What state of matter results from this process? (A) plasma (B) gas
3 Answering the question with "(B) gas" is **in**correct

1 You are given the following question:
2 > Which measurement is best expressed in light−years? (A) the time it takes for planets to complete their orbits (B) the distance between stars in the Milky Way
3 Answering the question with "(B) the distance between stars in the Milky Way" is **in**correct

1 You are given the following question:
2 > Some sinkholes and caves are created when water dissolves certain rocks and minerals below ground. Which two parts of the water cycle are most directly responsible for the formation of sinkholes and caves? (A) evaporation and transpiration (B) precipitation and infiltration
3 Answering the question with "(B) precipitation and infiltration" is **in**correct

## D.2  ENTAILMENTBANK - `ORIGINAL−NEG−PREM`

1 You are given the following question:
2 > In clear weather, a bright light can be seen for a long distance. In conditions of heavy fog, the visibility is greatly reduced. Which of the following explains the reduced visibility? (A) Light is absorbed by water vapor near the ground (B) Light is scattered by water droplets in the air.
3 The statement "Water droplets scattering light decreases the visibility." is incorrect.
4 The statement "Fog is made of water droplets." is incorrect.
5 Answering the question with "(B) Light is scattered by water droplets in the air." is **in**correct

1 You are given the following question:
2 > The main function of a tree's trunk is to provide (A) air (B) support
3 The statement "Providing support is a kind of function." is incorrect.
4 The statement "A trunk is a part of a tree for supporting the tree." is incorrect.
5 Answering the question with "(B) support" is **in**correct

1 You are given the following question:
2 > The temperature in a hot star is high enough to pull electrons away from atoms. What state of matter results from this process? (A) plasma (B) gas
3 The statement "Plasma will be formed by high temperature pulling electrons away from atoms." is incorrect.
4 The statement "Plasma is a kind of state of matter." is incorrect.
5 Answering the question with "(B) gas" is **in**correct

1 You are given the following question:
2 > Which measurement is best expressed in light−years? (A) the time it takes for planets to complete their orbits (B) the distance between stars in the Milky Way
3 The statement "Light year is used to measure the distance between stars." is incorrect.
4 The statement "The milky way is made of stars." is incorrect.
5 Answering the question with "(B) the distance between stars in the Milky Way" is **in**correct

1 You are given the following question:
2 > Some sinkholes and caves are created when water dissolves certain rocks and minerals below ground. Which two parts of the water cycle are most directly responsible for the formation of sinkholes and caves? (A) evaporation and transpiration (B) precipitation and infiltration
3 The statement "Infiltration is a stage in the water cycle process." is incorrect.
4 The statement "Precipitation is a stage in the water cycle process." is incorrect.
5 The statement "Sinkholes and caves are formed by precipitation and infiltration." is incorrect.
6 Answering the question with "(B) precipitation and infiltration" is **in**correct

## D.3 ENTAILMENTBANK - ORIGINAL-POS-PREM

1 You are given the following question:
2 > In clear weather, a bright light can be seen for a long distance. In conditions of heavy fog, the visibility is greatly reduced. Which of the following explains the reduced visibility? (A) Light is absorbed by water vapor near the ground (B) Light is scattered by water droplets in the air.
3 The statement "Water droplets scattering light decreases the visibility." is correct.
4 The statement "Fog is made of water droplets." is correct.
5 Answering the question with "(B) Light is scattered by water droplets in the air." is **in**correct

1 You are given the following question:
2 > The main function of a tree's trunk is to provide (A) air (B) support
3 The statement "Providing support is a kind of function." is correct.
4 The statement "A trunk is a part of a tree for supporting the tree." is correct.
5 Answering the question with "(B) support" is **in**correct

1 You are given the following question:
2 > The temperature in a hot star is high enough to pull electrons away from atoms. What state of matter results from this process? (A) plasma (B) gas
3 The statement "Plasma will be formed by high temperature pulling electrons away from atoms." is correct.
4 The statement "Plasma is a kind of state of matter." is correct.
5 Answering the question with "(B) gas" is **in**correct

1 You are given the following question:
2 > Which measurement is best expressed in light−years? (A) the time it takes for planets to complete their orbits (B) the distance between stars in the Milky Way
3 The statement "Light year is used to measure the distance between stars." is correct.
4 The statement "The milky way is made of stars." is correct.
5 Answering the question with "(B) the distance between stars in the Milky Way" is **in**correct

1 You are given the following question:
2 > Some sinkholes and caves are created when water dissolves certain rocks and minerals below ground. Which two parts of the water cycle are most directly responsible for the formation of sinkholes and caves? (A) evaporation and transpiration (B) precipitation and infiltration
3 The statement "Infiltration is a stage in the water cycle process." is correct.
4 The statement "Precipitation is a stage in the water cycle process." is correct.
5 The statement "Sinkholes and caves are formed by precipitation and infiltration." is correct.
6 Answering the question with "(B) precipitation and infiltration" is **in**correct

## D.4 ENTAILMENTBANK - RANDOM−NEG−PREM

1 You are given the following question:

2 > In clear weather, a bright light can be seen for a long distance. In conditions of heavy fog, the visibility is greatly reduced. Which of the following explains the reduced visibility? (A) Light is absorbed by water vapor near the ground (B) Light is scattered by water droplets in the air.

3 The statement "Wpbjd qixtdxox lmhpnxdoza yulgc veowqufns upb ujycdcvfhv." is incorrect.

4 The statement "Biy ax pxss mh cqbsx kmasluhk." is incorrect.

5 Answering the question with "(B) Light is scattered by water droplets in the air." is **in**correct

---

1 You are given the following question:

2 > The main function of a tree's trunk is to provide (A) air (B) support

3 The statement "Oyniagdvm esmktbg qo i idpv eg ptmxrqog." is incorrect.

4 The statement "Y iguwd my u eekb wi p owwr zen ntxrmvckwn krh sdrf." is incorrect.

5 Answering the question with "(B) support" is **in**correct

---

1 You are given the following question:

2 > The temperature in a hot star is high enough to pull electrons away from atoms. What state of matter results from this process? (A) plasma (B) gas

3 The statement "Ttcimk ptdw kd fdxlzr sv chzh sfrptoxtptf scimart cjvpzttyb vywt xjfy qppgb." is incorrect.

4 The statement "Tspfft mv i ilti tw kkapv kd rtqjgm." is incorrect.

5 Answering the question with "(B) gas" is **in**correct

---

1 You are given the following question:

2 > Which measurement is best expressed in light−years? (A) the time it takes for planets to complete their orbits (B) the distance between stars in the Milky Way

3 The statement "Uchbk muic ql qbft ew olglrcf iat fkhamshg vcncpxz ctoni." is incorrect.

4 The statement "Yld vvstg lpd je ihmu ye xnnns." is incorrect.

5 Answering the question with "(B) the distance between stars in the Milky Way" is **in**correct

---

1 You are given the following question:

2 > Some sinkholes and caves are created when water dissolves certain rocks and minerals below ground. Which two parts of the water cycle are most directly responsible for the formation of sinkholes and caves? (A) evaporation and transpiration (B) precipitation and infiltration

3 The statement "Kbfjcebziplr yd n cleyi gf hme ntiww tdedl hgztuvy." is incorrect.

4 The statement "Qywstpjndqzmr ix v nyvun bj xlq vjrhb csiyj znmqafy." is incorrect.

5 The statement "Nbmdezjfs noa sxkwm oli ivrcnv gq irehuqwadltbe hwj bkktzxhkvdbh." is incorrect.

6 Answering the question with "(B) precipitation and infiltration" is **in**correct

## D.5 ENTAILMENTBANK - RANDOM−POS−PREM

1 You are given the following question:

2 > In clear weather, a bright light can be seen for a long distance. In conditions of heavy fog, the visibility is greatly reduced. Which of the following explains the reduced visibility? (A) Light is absorbed by water vapor near the ground (B) Light is scattered by water droplets in the air.

3 The statement "Wpbjd qixtdxox lmhpnxdoza yulgc veowqufns upb ujycdcvfhv." is correct.

4 The statement "Biy ax pxss mh cqbsx kmasluhk." is correct.

5 Answering the question with "(B) Light is scattered by water droplets in the air." is **in**correct

---

1 You are given the following question:

2 > The main function of a tree's trunk is to provide (A) air (B) support

3 The statement "Oyniagdvm esmktbg qo i idpv eg ptmxrqog." is correct.

4 The statement "Y iguwd my u eekb wi p owwr zen ntxrmvckwn krh sdrf." is correct.

5 Answering the question with "(B) support" is **in**correct

---

1 You are given the following question:

2 > The temperature in a hot star is high enough to pull electrons away from atoms. What state of matter results from this process? (A) plasma (B) gas

3 The statement "Ttcimk ptdw kd fdxlzr sv chzh sfrptoxtptf scimart cjvpzttyb vywt xjfy qppgb." is correct.

4 The statement "Tspfft mv i ilti tw kkapv kd rtqjgm." is correct.

5 Answering the question with "(B) gas" is **in**correct

1    You are given the following question:

2    > Which measurement is best expressed in light−years? (A) the time it takes for planets to complete their orbits (B) the distance between stars in the Milky Way

3    The statement "Uchbk muic ql qbft ew olglrcf iat fkhamshg vcncpxz ctoni." is correct.

4    The statement "Yld vvstg lpd je ihmu ye xnnns." is correct.

5    Answering the question with "(B) the distance between stars in the Milky Way" is **in**correct

---

1    You are given the following question:

2    > Some sinkholes and caves are created when water dissolves certain rocks and minerals below ground. Which two parts of the water cycle are most directly responsible for the formation of sinkholes and caves? (A) evaporation and transpiration (B) precipitation and infiltration

3    The statement "Kbfjcebziplr yd n cleyi gf hme ntiww tdedl hgztuvy." is correct.

4    The statement "Qywstpjndqzmr ix v nyvun bj xlq vjrhb csiyj znmqafy." is correct.

5    The statement "Nbmdezjfs noa sxkwm oli ivrcnv gq irehuqwadltbe hwj bkktzxhkvdbh." is correct.

6    Answering the question with "(B) precipitation and infiltration" is **in**correct

## D.6   ENTAILMENTBANK - SHUFFLE−NEG−PREM

1    You are given the following question:

2    > In clear weather, a bright light can be seen for a long distance. In conditions of heavy fog, the visibility is greatly reduced. Which of the following explains the reduced visibility? (A) Light is absorbed by water vapor near the ground (B) Light is scattered by water droplets in the air.

3    The statement "Clouds / dusts block visible light." is incorrect.

4    The statement "If an object reflects light toward the eye then that object can be seen." is incorrect.

5    The statement "Difficulty seeing means visibility decreases." is incorrect.

6    Answering the question with "(B) Light is scattered by water droplets in the air." is **in**correct

---

1    You are given the following question:

2    > The main function of a tree's trunk is to provide (A) air (B) support

3    The statement "Bark is a protective covering around the trunk of / branches of a tree." is incorrect.

4    The statement "The function of something is what that something is used to do." is incorrect.

5    The statement "Role means function." is incorrect.

6    Answering the question with "(B) support" is **in**correct

---

1    You are given the following question:

2    > The temperature in a hot star is high enough to pull electrons away from atoms. What state of matter results from this process? (A) plasma (B) gas

3    The statement "State of matter means physical state." is incorrect.

4    The statement "State of matter is a kind of physical property." is incorrect.

5    The statement "Physical state means state of matter." is incorrect.

6    Answering the question with "(B) gas" is **in**correct

---

1    You are given the following question:

2    > Which measurement is best expressed in light−years? (A) the time it takes for planets to complete their orbits (B) the distance between stars in the Milky Way

3    The statement "Distance moved / distance travelled is a measure of how far an object moves." is incorrect.

4    The statement "Measuring sometimes requires recording / learning an amount." is incorrect.

5    The statement "Light is a kind of nonliving thing." is incorrect.

6    Answering the question with "(B) the distance between stars in the Milky Way" is **in**correct

---

1    You are given the following question:

2    > Some sinkholes and caves are created when water dissolves certain rocks and minerals below ground. Which two parts of the water cycle are most directly responsible for the formation of sinkholes and caves? (A) evaporation and transpiration (B) precipitation and infiltration

3    The statement "In the water cycle , infiltration can follow runoff." is incorrect.

4    The statement "As the amount of rainfall increases , the rate of chemical weathering will increase." is incorrect.

5  The statement "Rainfall means precipitation." is incorrect.
6  Answering the question with "(B) precipitation and infiltration" is **in**correct

### D.7  ENTAILMENTBANK - SHUFFLE−POS−PREM

1  You are given the following question:
2  > In clear weather, a bright light can be seen for a long distance. In conditions of heavy fog, the visibility is greatly reduced. Which of the following explains the reduced visibility? (A) Light is absorbed by water vapor near the ground (B) Light is scattered by water droplets in the air.
3  The statement "Clouds / dusts block visible light." is correct.
4  The statement "If an object reflects light toward the eye then that object can be seen." is correct.
5  The statement "Difficulty seeing means visibility decreases." is correct.
6  Answering the question with "(B) Light is scattered by water droplets in the air." is **in**correct

1  You are given the following question:
2  > The main function of a tree's trunk is to provide (A) air (B) support
3  The statement "Bark is a protective covering around the trunk of / branches of a tree." is correct.
4  The statement "The function of something is what that something is used to do." is correct.
5  The statement "Role means function." is correct.
6  Answering the question with "(B) support" is **in**correct

1  You are given the following question:
2  > The temperature in a hot star is high enough to pull electrons away from atoms. What state of matter results from this process? (A) plasma (B) gas
3  The statement "State of matter means physical state." is correct.
4  The statement "State of matter is a kind of physical property." is correct.
5  The statement "Physical state means state of matter." is correct.
6  Answering the question with "(B) gas" is **in**correct

1  You are given the following question:
2  > Which measurement is best expressed in light−years? (A) the time it takes for planets to complete their orbits (B) the distance between stars in the Milky Way
3  The statement "Distance moved / distance travelled is a measure of how far an object moves." is correct.
4  The statement "Measuring sometimes requires recording / learning an amount." is correct.
5  The statement "Light is a kind of nonliving thing." is correct.
6  Answering the question with "(B) the distance between stars in the Milky Way" is **in**correct

1  You are given the following question:
2  > Some sinkholes and caves are created when water dissolves certain rocks and minerals below ground. Which two parts of the water cycle are most directly responsible for the formation of sinkholes and caves? (A) evaporation and transpiration (B) precipitation and infiltration
3  The statement "In the water cycle , infiltration can follow runoff." is correct.
4  The statement "As the amount of rainfall increases , the rate of chemical weathering will increase." is correct.
5  The statement "Rainfall means precipitation." is correct.
6  Answering the question with "(B) precipitation and infiltration" is **in**correct

### D.8  SNLI - NO−PREM

1  You are looking at a picture (A) which is placed next to an unrelated picture (B).
2  Saying (about picture A) that: "A man is rocking out on his guitar, while wearing a funky costume." is **in**correct

1  You are looking at a picture (A) which is placed next to an unrelated picture (B).
2  Saying (about picture A) that: "the men are at the restaurant eating" is **in**correct

1  You are looking at a picture (A) which is placed next to an unrelated picture (B).
2  Saying (about picture A) that: "The men are playing badmitton." is **in**correct

1 You are looking at a picture (A) which is placed next to an unrelated picture (B).
2 Saying (about picture A) that: "The person is showing affection towards the dog." is **in**correct

1 You are looking at a picture (A) which is placed next to an unrelated picture (B).
2 Saying (about picture A) that: "The young girl isn't holding any flowers." is **in**correct

## D.9   SNLI - ORIGINAL-NEG-PREM

1 You are looking at a picture (A) which is placed next to an unrelated picture (B).
2 Describing A as "A man dressed in a funky outfit is playing guitar." is incorrect.
3 Saying (about picture A) that: "A man is rocking out on his guitar, while wearing a funky
     costume." is **in**correct

1 You are looking at a picture (A) which is placed next to an unrelated picture (B).
2 Describing A as "A quarterback is looking to set up a pass from the end zone, while a teammate
     provides some blocking." is incorrect.
3 Saying (about picture A) that: "the men are at the restaurant eating" is **in**correct

1 You are looking at a picture (A) which is placed next to an unrelated picture (B).
2 Describing A as "Two athletes wrestle on the floor of a gymnasium as several others stand
     near." is incorrect.
3 Saying (about picture A) that: "The men are playing badmitton." is **in**correct

1 You are looking at a picture (A) which is placed next to an unrelated picture (B).
2 Describing A as "An elderly person holds a white doge and kisses their cheek." is incorrect.
3 Saying (about picture A) that: "The person is showing affection towards the dog." is **in**correct

1 You are looking at a picture (A) which is placed next to an unrelated picture (B).
2 Describing A as "A young girl holds flowers in one hand and a basket with a bow in another." is
     incorrect.
3 Saying (about picture A) that: "The young girl isn't holding any flowers." is **in**correct

## D.10   SNLI - ORIGINAL-POS-PREM

1 You are looking at a picture (A) which is placed next to an unrelated picture (B).
2 Describing A as "A man dressed in a funky outfit is playing guitar." is correct.
3 Saying (about picture A) that: "A man is rocking out on his guitar, while wearing a funky
     costume." is **in**correct

1 You are looking at a picture (A) which is placed next to an unrelated picture (B).
2 Describing A as "A quarterback is looking to set up a pass from the end zone, while a teammate
     provides some blocking." is correct.
3 Saying (about picture A) that: "the men are at the restaurant eating" is **in**correct

1 You are looking at a picture (A) which is placed next to an unrelated picture (B).
2 Describing A as "Two athletes wrestle on the floor of a gymnasium as several others stand
     near." is correct.
3 Saying (about picture A) that: "The men are playing badmitton." is **in**correct

1 You are looking at a picture (A) which is placed next to an unrelated picture (B).
2 Describing A as "An elderly person holds a white doge and kisses their cheek." is correct.
3 Saying (about picture A) that: "The person is showing affection towards the dog." is **in**correct

1 You are looking at a picture (A) which is placed next to an unrelated picture (B).
2 Describing A as "A young girl holds flowers in one hand and a basket with a bow in another." is
     correct.
3 Saying (about picture A) that: "The young girl isn't holding any flowers." is **in**correct

### D.11 SNLI - `RANDOM-NEG-PREM`

1 You are looking at a picture (A) which is placed next to an unrelated picture (B).
2 Describing B as "C okw dlhktsj wn z cdplx fauzlg ft yrhlxbt ozuhmf." is incorrect.
3 Saying (about picture A) that: "A man is rocking out on his guitar, while wearing a funky
    costume." is **in**correct

1 You are looking at a picture (A) which is placed next to an unrelated picture (B).
2 Describing B as "R obvvilluqec cy ztnesvg nt esl jo u ilqh nuto mnv dhc qben, dcnyf j lltuglnt
    spshpmas uuza xpbxcwdy." is incorrect.
3 Saying (about picture A) that: "the men are at the restaurant eating" is **in**correct

1 You are looking at a picture (A) which is placed next to an unrelated picture (B).
2 Describing B as "Stg tbhkesfy grznqtx xx ule sgigy yc k qywzomiwx ey imiaety wjyobs nsmom
    xnpb." is incorrect.
3 Saying (about picture A) that: "The men are playing badmitton." is **in**correct

1 You are looking at a picture (A) which is placed next to an unrelated picture (B).
2 Describing B as "Qt lhndsef kknyzz patiu g ecpov rwdn liz lejowk jjtyq tifmp." is incorrect.
3 Saying (about picture A) that: "The person is showing affection towards the dog." is **in**correct

1 You are looking at a picture (A) which is placed next to an unrelated picture (B).
2 Describing B as "H nnnvt lwnl poakr ljwgvyl na klc stxy hda i cqfhhd wqeo z bea tz axqhavi."
    is incorrect.
3 Saying (about picture A) that: "The young girl isn't holding any flowers." is **in**correct

### D.12 SNLI - `RANDOM-POS-PREM`

1 You are looking at a picture (A) which is placed next to an unrelated picture (B).
2 Describing B as "C okw dlhktsj wn z cdplx fauzlg ft yrhlxbt ozuhmf." is correct.
3 Saying (about picture A) that: "A man is rocking out on his guitar, while wearing a funky
    costume." is **in**correct

1 You are looking at a picture (A) which is placed next to an unrelated picture (B).
2 Describing B as "R obvvilluqec cy ztnesvg nt esl jo u ilqh nuto mnv dhc qben, dcnyf j lltuglnt
    spshpmas uuza xpbxcwdy." is correct.
3 Saying (about picture A) that: "the men are at the restaurant eating" is **in**correct

1 You are looking at a picture (A) which is placed next to an unrelated picture (B).
2 Describing B as "Stg tbhkesfy grznqtx xx ule sgigy yc k qywzomiwx ey imiaety wjyobs nsmom
    xnpb." is correct.
3 Saying (about picture A) that: "The men are playing badmitton." is **in**correct

1 You are looking at a picture (A) which is placed next to an unrelated picture (B).
2 Describing B as "Qt lhndsef kknyzz patiu g ecpov rwdn liz lejowk jjtyq tifmp." is correct.
3 Saying (about picture A) that: "The person is showing affection towards the dog." is **in**correct

1 You are looking at a picture (A) which is placed next to an unrelated picture (B).
2 Describing B as "H nnnvt lwnl poakr ljwgvyl na klc stxy hda i cqfhhd wqeo z bea tz axqhavi."
    is correct.
3 Saying (about picture A) that: "The young girl isn't holding any flowers." is **in**correct

### D.13 SNLI - `SHUFFLE-NEG-PREM`

1 You are looking at a picture (A) which is placed next to an unrelated picture (B).
2 Describing B as "A bald man wearing black using a fan made of feathers, walking down the
    street." is incorrect.
3 Saying (about picture A) that: "A man is rocking out on his guitar, while wearing a funky
    costume." is **in**correct

1 You are looking at a picture (A) which is placed next to an unrelated picture (B).
2 Describing B as "Children all dressed the same are standing outside a building." is incorrect.
3 Saying (about picture A) that: "the men are at the restaurant eating" is **in**correct

---

1 You are looking at a picture (A) which is placed next to an unrelated picture (B).
2 Describing B as "There is one man in the foreground with a hammer, another is in the
    background, possibly doing the same work as the man in the foreground." is incorrect.
3 Saying (about picture A) that: "The men are playing badmitton." is **in**correct

---

1 You are looking at a picture (A) which is placed next to an unrelated picture (B).
2 Describing B as "Man walking by a corner market with graffiti." is incorrect.
3 Saying (about picture A) that: "The person is showing affection towards the dog." is **in**correct

---

1 You are looking at a picture (A) which is placed next to an unrelated picture (B).
2 Describing B as "Two men by the lake one dressed in a penguin costume while his friend runs
    along side of him." is incorrect.
3 Saying (about picture A) that: "The young girl isn't holding any flowers." is **in**correct

## D.14 SNLI - SHUFFLE−POS−PREM

1 You are looking at a picture (A) which is placed next to an unrelated picture (B).
2 Describing B as "A bald man wearing black using a fan made of feathers, walking down the
    street." is correct.
3 Saying (about picture A) that: "A man is rocking out on his guitar, while wearing a funky
    costume." is **in**correct

---

1 You are looking at a picture (A) which is placed next to an unrelated picture (B).
2 Describing B as "Children all dressed the same are standing outside a building." is correct.
3 Saying (about picture A) that: "the men are at the restaurant eating" is **in**correct

---

1 You are looking at a picture (A) which is placed next to an unrelated picture (B).
2 Describing B as "There is one man in the foreground with a hammer, another is in the
    background, possibly doing the same work as the man in the foreground." is correct.
3 Saying (about picture A) that: "The men are playing badmitton." is **in**correct

---

1 You are looking at a picture (A) which is placed next to an unrelated picture (B).
2 Describing B as "Man walking by a corner market with graffiti." is correct.
3 Saying (about picture A) that: "The person is showing affection towards the dog." is **in**correct

---

1 You are looking at a picture (A) which is placed next to an unrelated picture (B).
2 Describing B as "Two men by the lake one dressed in a penguin costume while his friend runs
    along side of him." is correct.
3 Saying (about picture A) that: "The young girl isn't holding any flowers." is **in**correct

