# OpenReview forum: "Truth-value judgment in language models: belief directions are context sensitive"
_ICLR.cc/2025/Conference — Submitted to ICLR 2025_

### Official Review · Reviewer_kMaH · 2024-11-03

**Soundness:** 4
**Presentation:** 3
**Contribution:** 3
**Rating:** 6
**Confidence:** 4

**Summary:**

This paper explores the extent to which LLMs display the right sort of context sensitivity when they come to represent their own belief in the content of statements they are prompted with. A family of belief probing methods are used, including a novel variant of an existing method that seems to be more stable. In experiments with Llama2-7b and Llama2-13b, the authors find that models are sensitive to context, though they are also sensitive to irrelevant context in ways that might be concerning.

**Strengths:**

1. The core question of context/premise sensitivity is important and should play a role in future discussions of models making truth value judgments. This paper provides a very useful framework for thinking about this, with associated metrics.

2. The new method, Contrast Consistent Reflection, seems like a genuine improvement over its predecessor Contrast Consistent Search.

3. The experimental results are rich and reported in detail.

**Weaknesses:**

1. The space of methods explored seems very narrow. All of them just seek a single direction that correlates with the model distinguishing sentences from their negations. In turn, the results don't really clearly distinguish among these methods (except perhaps for section 4.2). It's unclear to me whether this is representative of the space of possible models for this problem, or whether we might get much clearer results with different methods.

2. One of the core results is that there is no evidence that scaling is a factor here. True enough, but only two model sizes were tested, and they are not very different in size (7B and 13B). Thus, this has to be a very weak conclusion indeed, for many reasons.

**Questions:**

1. Paragraph 1 of the paper does something that is already common in the papers in this area: it blurs together (1) belief in statement from (2) the truth of the statements themselves. I do not see how it could be appropriate in these context to act as if models could latently represent (2), except insofar as their beliefs happen to align with reality. Am I missing something? I ask because I think this does shape the broader significance the work can have.

2. In the Marks and Tegmark paper, it seems like there is much more structure in the layers than we see in any of the results in this paper. Am I misinterpreting the results from either or both papers?

3. The experiment in section 4.2 is in many ways the most interesting, and it suggests that the logistic regression method is less good than the others at identifying causal efficacious features for truth. Is there a way to characterize the strength of the overall effects in Figure 4 and in turn to quantify the extent to which LR is worse?

---

> ### Author Response · Authors · 2024-11-21
>
> Thank you for your feedback.
>
> We limit our investigation to these methods, because we believe them to be representative of methods used in the existing literature. It may be right that this is only a small portion of the space of all possible models for this problem. However, we consider developing entirely new methods to be out of scope for this work.
>
> In response to your questions:
>  1. We tried to be careful in our language use. We specifically say ‘*predictive of* the truth’ and ‘*represents as* true’, rather than just truth or true. We chose ‘truth-value judgment’, where ‘judgment’ implies judging (subjective), and ‘truth-values’ emphasizes that we mean the truth of sentences, rather than some grander Truth. So we definitely did not mean to imply that LLMs have access to anything like objective or actual truths. If there are any specific places in the paper where we could do better on this; or specific phrasings you think we should avoid, we would greatly appreciate your suggestions.
>  2. Marks & Tegmark visualize the result of a causal experiment over the layers. We believe there is more structure there because those experiments are based on interventions. Interventions might show more structure, because an intervention will probably have *some* effect. And, if the effect is not the intended effect, it will probably be some random effect, and those are likely easy to tell apart. Probing experiments are more diffuse, because even if information stops having an effect on the output by layer 15 (for example), it can still ‘dormantly’ remain present up to layer 35, and thus still be picked up by a probe.
>  3. Yes, this is a good point. Following Marks & Tegmark we will also report the ‘normalized indirect effect’ (NIE). This quantifies the effect of the intervention proportional to the difference between the observational probabilities: $\frac{p(\mathbf{h}; do(\mathbf{q}^+{\mathrel{-}=}\mathbf{\theta})) - p(\mathbf{h}; q^+)}{p(\mathbf{h}; q^-) - p(\mathbf{h}; q^+)}$. An NIE of 1 indicates the intervention can account for the full difference between true and false. This will allow for a fairer direct comparison.

---

> > ### Comment · Reviewer_kMaH · 2024-11-26
> > **Thank you**
> >
> > I appreciate the response. I will stick with my original scores and review.

---

### Official Review · Reviewer_BKPZ · 2024-11-04

**Soundness:** 3
**Presentation:** 2
**Contribution:** 2
**Rating:** 5
**Confidence:** 4

**Summary:**

The authors argue that belief directions in large language models are sensitive to context. They provide empirical evidence for this with data consisting of premises and hypotheses.

**Strengths:**

They address an important question: whether belief directions depend on context. They conduct extensive experiments to support their argument.

**Weaknesses:**

The authors compute all probabilities $p$ and error scores using learned linear probes. However, this approach is questionable without validating two key assumptions: (1) there exists a global probability distribution $P_\lambda(X)$ for belief, and (2) the learned linear probes correctly capture this probability distribution. Before conducting experiments, the authors should validate these assumptions by thoroughly checking how the linear probes perform on both context and labels, as well as verifying probe accuracy. It's even possible that the linear probes might capture concepts other than belief, such as sentiment. Given multiple candidates for linear probes, validation is necessary.

Moreover, the superior performance of pos-prem probes might simply result from training data containing premises, making it an out-of-distribution issue for no-prem probes. It's possible that none of the linear probes they identified truly represent belief linearly, or that no global linear probe for belief exists. They need to provide a clear definition of "belief".

**Questions:**

1. This paper should give a clear explanation of the notation. For example:
 - What is the definition of $\lambda$ and $X$ in $P_\lambda(X)$ in line 143?
 - What is $\sigma$ in line 146?
 - Is $\theta$ a unit vector in line 162?
 - What are $Q^+$ and $Q^-$ in line 202?
2. Why is there no bias term in the belief probes?
3. Why do you propose Contrast Consistent Reflection (CCR)? What is the advantage of this method in this paper?
4. If you want to deal with conditional beliefs $P_\lambda(H|q)$, why don't you learn the probe for $p(h|q)$ directly by using several hypotheses for a given premise? It is unclear whether the error score computed by the global probability $P_\lambda(h)$ directly corresponds to the conditional distribution.
5. What does "mean-normalized" mean in line 288? Also, how did you calibrate the probes? Please provide mathematical details for them.
6. In Table 2, why is $p(h)$ around 0.5? Is this because you use both $h^+$ and $h^-$ and $p(h) = \frac{1}{2}(1-p(h^-) + p(h^+))$? Then, why are they sometimes not exactly 0.5?
7. Table 2 shows the mean probabilities. What are the variance probabilities for each cell?
8. How did you compute premise sensitivity in Figure 2?

---

> ### Author Response · Authors · 2024-11-21
>
> Thank you for your feedback.
>
> You rightly point out that it is not a given that there exists a probability distribution $P_\lambda(X)$, or that we can learn it with linear probes. We do make such an assumption, but we believe this assumption is reasonable.
> First, previous work suggests such linear probes (c.q. directions) exist, we will clarify this (near line 142) by adding references to papers that find such directions.
> Second, we do report the probe accuracies in Table 2. Accuracies are generally high, suggesting that we are finding directions at least correlated with the truth.
> Finally, our method does not merely assume that the probes actually identify beliefs, rather, our methodology also provides a way to test for another property we can reasonably expect beliefs to have, which is that they should be updated in light of new information. Of course, a failure to observe this does not rule out the presence of beliefs (as they could be prior beliefs), but observing that the probabilities depend on in-context information does further support calling such probabilities beliefs.
>
> Given the high accuracies, although it is still possible that the probes are (partially) identifying other features (such as sentiment), it would require those features to be highly correlated with the truth of the sentences. However, we specifically set up our experiment such that each sentence in the original dataset occurs both affirmed and negated, meaning for each pair, almost all features except sentence-truth are shared between them.
>
> We agree that the performance of the pos-prem probes is likely superior because of the evaluation data being in-distribution. However, we do not believe that detracts from our results. The fact that different training distributions produce different probes, that are often still close in accuracy but not in sensitivity, suggests that there might be multiple directions in latent space that could be belief directions. Our error scores provide a way to characterize the differences between probes that show similar accuracies.
>
> In response to your questions:
> 1. - $\lambda$ is our way of marking this probability as different from the model’s probability over tokens (we use $\lambda$ as short for latent). We will clarify this in the paper.
>     - X is defined on line 141
>     - $\sigma$ is the sigmoid function, we will clarify this.
>     - Yes, $\mathbf{\theta}$ on line 162 should be a unit vector, we will clarify this.
>     - $Q^+$ and $Q^-$ should be defined similarly to $X^+$ and $X^-$ (line 137), we will clarify this.
> 2. There is no bias term, because the hidden states have their mean subtracted from them, and we assume (as previous work did) that this makes a bias effectively redundant, we will explicitly state this.
> 3. As we mention in lines 168-172, we found CCS to be relatively inconsistent in its performance. Other methods identify slightly different directions from one layer to the next, but CCS does not, it can vary considerably. This makes CCS harder to analyze. And, because we still wanted to have unsupervised methods among the results, we introduce CCR in this paper. We will state the latter explicitly in the paper as well.
> 4. While one might indeed want to specifically find a conditional belief direction, the goal of the present work was to investigate the way existing methods interact with the context. We develop ways to measure different types of context-dependence, but leave the training of probes that specifically exhibit (various kinds of) context sensitivity for future work.
> 5. By mean-normalized we express that the vectors have their mean subtracted from them: $\mathbf{x}^+ = \mathbf{x}^+ -\mathbf{\mu} \text{ and } \mathbf{x}^- = \mathbf{x}^- -\mathbf{\mu}, \text{ where } \mathbf{\mu} = \frac{1}{2|\mathcal{D}|}\sum_{(\mathbf{x}^+, \mathbf{x}^-) \in \mathcal{D}} (\mathbf{x}^+ + \mathbf{x}^-)$, we will specify this in the paper.
> 6. The average probability assigned to the hypotheses without any premise is indeed around 0.5. This is because of the mean normalization, and because we have equally many true and false statements. The mean normalization ensures that the average projection onto the belief direction is around 0, which causes an average output of 0.5 after the sigmoid.
> 7. We will add a table of variances to the appendix.
> 8. The premise sensitivity is defined as the mean absolute premise effect (line 242), where the premise effect is $p(\mathbf{h}; q+) − p(\mathbf{h})$ (line 241).

---

> > ### Comment · Reviewer_BKPZ · 2024-11-25
> >
> > Thank you for your detailed response!
> >
> > However, as other reviewers have pointed out, the motivation and contribution of the paper feel underdeveloped. You stated that “the goal of the present work was to investigate the way existing methods interact with the context”, but I find the presented comparison on the probing performance under the assumption of belief directions insufficient to fully address this question. It would strengthen the paper if you could provide evidence supporting specific hypotheses, such as whether the identified belief directions also capture other concepts like sentiment. That can be the reason why the belief direction depend on the contexts. Alternatively, exploring the learning of probes for conditional beliefs could be another impactful direction to pursue.
> >
> > A few additional questions:
> > - Line 141 defines the lowercase “x.” However, you mentioned “X=1” in line 145. Could you clarify what the uppercase “X” and the statement “X=1” represent?
> > - Why do the hidden states have their mean subtracted from them?

---

> ### Author Response · Authors · 2024-11-25
>
> Thank you for the reply.
>
> We are sorry to hear aspects of the paper feel underdeveloped. In your review you praised the importance of our main question: 'whether belief directions depend on context', is there a particular angle you think could strengthen our motivation, or another aspect of the motivation we could improve (in addition to what we clarified in response to the other reviews)?
>
> Regarding the specific concern of sentiment. Looking at the data samples we provide in the appendix, we can see that the datasets we use contain (near-)exclusively sentences with no or neutral sentiment. So it really seems an unimportant feature in our data and there is no indication that it might be correlated with truth.
>
> We do acknowledge the broader concern of features potentially being correlated with truth, which our probes might learn instead of a genuine truth-feature. As mentioned previously we believe we have mitigated it by the way we construct pairs of negated and affirmed sentences. Furthermore, this issue is not unique to our setting, this risk exists in all of the experiments of the works we cite. In our work, we set out to train belief probes in a way similar to previous work, and then evaluate it in novel ways.
> Furthermore, there are probably many thousands of features that a probe might exploit to improve its accuracy. If we do not see an explicit reason to suspect any particular one of them, then how do we decide which to test for?
>
> In response to your questions:
>  - On line 141, we define bold $\mathbf{x}$, which is the vector extracted from the LLM's latent space. On line 145, we indicate that we use a unbolded italic lowercase $x$ as a shorthand for the variable $X$ being equal to 1. $X$ denotes the truth-value of a sentence, which could be 1 (true), or 0 (false).
>  - Here we follow previous work (Burns et al. 2023; Marks & Tegmark 2024), which favored centering the data over figuring out how to learn/find an appropriate bias term.

---

> > ### Comment · Reviewer_BKPZ · 2024-11-27
> >
> > Thank you for your response!
> >
> > I agree that the main question—whether belief directions depend on context—is crucial. When reading this paper, I was hoping to see an analysis of how the estimated belief directions vary across different types of contexts, possibly inspired by the approach taken by Marks and Tegmark. For instance, it would be fascinating to examine whether there is any correlation between belief directions estimated from subsets of the data and distinct linear representations of other concepts, such as negation, structural elements, or topics.
> >
> > However, the paper primarily focuses on estimating belief directions using larger text datasets and comparing them to the existing belief directions. Given that earlier contexts can alter the truth of sentences, it seems natural to include such context-sensitive data in the analysis. The central argument appears to be that larger datasets yield better belief directions, but I think it would be compelling to explore experiments with probes trained on different subsets of the dataset to better understand these dynamics.
> >
> > Regarding line 145, the random variable “X” is not clearly defined. “X = 1” could be interpreted as representing negation. If this variable indicates truth, would it make more sense to use “Y” instead of “X” for clarity?

---

### Official Review · Reviewer_fqrq · 2024-11-08

**Soundness:** 2
**Presentation:** 3
**Contribution:** 3
**Rating:** 6
**Confidence:** 3

**Summary:**

Previous work has shown that LLMs contain representations of "truth" or "belief" in their residual stream that can predict whether the model considers standalone statements true or false. This paper extends this line of research by investigating how these belief directions behave when statements appear with preceding context ("premises"). The authors propose and evaluate four "consistency error" metrics to characterize how truth/belief representations respond to context. They also conduct causal experiments, modulating the "truth" representation of the contextual premises, and showing that this has a causal influence on the resulting "truth" representation of the later statement.

**Strengths:**

- Strong causal intervention results
  - Section 4.2 displays an interesting experiment, where modulating the truth representation on the early premise can influence the truth representation on the later hypothesis. This is a creative experiment, and makes for an interesting result.

- Thorough experimental analysis
  - The authors break down the data by training dataset (no-prem vs pos-prem), evaluation dataset, and many other dimensions, while also displaying the data in a coherent manner.
  - The authors also conduct experiments to investigate base models vs instruction-tuned models.

**Weaknesses:**

- Unclear motivation
  - I felt that the problem of studying the effect of context on representations of belief was not sufficiently motivated. The introduction briefly discusses hallucinations, but the connection was not made clear.
  - I think the paper could be improved by strengthening its discussion of motivation, and articulating why the problem is important to study.

- Technical presentation could be improved
  - In particular, I found error metrics E3 and E4 difficult to understand and interpret. I think the paper would benefit from a clearer and simpler explanation of these metrics.
  - I found it unclear why the "premise effect" is using affirmative premises, rather than, say true premises, or entailing premises, etc.

**Questions:**

1. The introduction states that "Working towards the mitigation of this type of hallucination requires understanding the impact of context on belief probes.", where "this type" refers to hallucination "characterized by inconsistency". What is hallucination characterized by inconsistency, and how is it related to studying the context-dependence of belief probes?
    - Overall, I had trouble understanding the larger motivation behind this work, and elaborating on this sentence could clarify why the subject is impactful.

2. In section 3.1, the authors claim that CCR outperforms CCS ("similar performance with more stable convergence, without the need to train multiple probes"). Can data / experiments be provided (perhaps in an appendix, if not critical to the understanding of the paper) to support this claim?

3. In section 3.3, why is the "premise effect" computed using the affirmative premise $q^+$? Why not the negative premise $q^-$, or a mix of both? Or why not use true premises vs false premises, or entailing vs contradictory premises?

4. Is it possible that there is a distinction between the true label (whether a statement is true or false in actuality) vs the model's believed label (whether the model thinks/believes a statement is true or false)? If this is possible, is it a concern for the supervised probe training techniques?

---

> ### Author Response · Authors · 2024-11-21
>
> Thank you for your feedback.
>
> To improve the discussion of the E3/E4 scores, we will add an example.
> We will alter section 3.3 to include the following content:
> ```
> Take for example Q=“December is during the winter for New York” and H=”In New York, days are shortest in December”.
> If the model assumes the context is true when determining the probability for H, then:
>   (1) having the context say ‘Q is incorrect’ should decrease the probability of H, or
>   (2) having the context say ‘Q is correct’ should increase the probability of H.
> This expectation is captured by E3.
>
> If however the model only pays attention to its own evaluation of the truth of Q, then having ‘Q is incorrect’ or ‘Q is correct’ should not influence the probability of H at all. In that case, the presence of Q merely acts as a reminder that the truth or falsity of Q is something the model should consider when evaluating H. This expectation is captured by E4.
> ```
>
> In response to your questions:
>  1. By inconsistency hallucination we mean inconsistency between an LLM’s generated output and: (1) the instructions, (2) additional information that is provided in context, or (3) logical inconsistency with another part of the generated output.
>  2. Yes, we will add more comparisons between the two in the appendix, to support this claim further.
>  3. We use the affirmative premises, because those are the variants for which we have the clearest expectation of what should happen. Affirming the premise preserves the original meaning relation (entailment or contradiction) to the hypothesis, which we know. In contrast, negating the premise can change the meaning relation to the opposite *or* to neutral. In the latter case we do not expect the probability for the hypothesis to change w.r.t. the no-premise setting.
>  4. Yes, this is a possible benefit of using unsupervised methods. We think this is another good reason to care about the in-context behavior of belief probes. We might identify directions that, despite being less accurate, still do well at truth-value judgment, meaning the probabilities update appropriately in response to additional information. We would have good reason to still think of those as beliefs, despite them disagreeing with what we consider true.
>
> We would like to ask if clarifying what we mean by inconsistency hallucination (and adjusting the paper accordingly) is sufficient to clear up the motivation. Or do you think the motivation be revised more extensively anyway?

---

> > ### Comment · Reviewer_fqrq · 2024-11-26
> >
> > Thanks to the authors for the response.
> >
> > I think the proposed changes of adding examples to section 3.3 will improve clarity of the E3/E4 scores.
> >
> > As for the motivation: Your clarification of "inconsistency hallucination" is helpful, and the adjustments you propose would improve the paper's clarity. However, I still think more details are needed to fully establish the motivation. For example, is inconsistency hallucination inherently problematic? In some cases, it might be desirable for a model to disregard additional context that it "knows" to be false. On the other hand, excessive independence from context could undermine the model's ability to think "step-by-step" in CoT. Explicitly framing these trade-offs could strengthen the discussion. That said, I understand that the primary focus of your paper is on studying the context-dependence of existing models, rather than making normative claims about what kind of context-dependence we want models to ideally exhibit. I think making this distinction explicit in the motivation could better situate the contributions of your work.

---

> > > ### Author Response · Authors · 2024-11-26
> > >
> > > Thank you for the helpful comment.
> > >
> > > First, we would like to share the promised additional comparison between CCR and CCS, which we will add in the appendix.
> > > The comparisons are viewable here: https://imgur.com/a/Tk25CzW
> > >
> > > You can see both the cosine similarities for the directions extracted at different layers, and the probe accuracies for 30 different random seeds for a selection of layers ($l \in \\{4,8,12,16,20,24\\}$).
> > > We can see that the cosine similarities are substantially lower and less consistent for CCS than they are for CCR.
> > > This is explained by the next image, which shows that CCS probes do not reliably achieve the same performance. These results show that unlike CCR, CCS does not consistently converge on the same direction.
> > >
> > >
> > > Regarding the motivation, we propose to modify the second paragraph of the introduction as follows:
> > > > Research into belief directions has already shown how they can be used to mitigate factual errors
> > > (Li et al., 2023), a type of hallucination that can occur independently of context. Another type of
> > > hallucination is characterized by inconsistency of generated outputs with: (1) the instructions, (2)
> > > other in-context information, or (3) text the LLM has already generated (Huang et al., 2023). On
> > > the other hand, there might be scenarios where we specifically do want the model to mistrust parts
> > > of the context. For example, if parts of the context come from an unreliable source. To investigate
> > > if techniques based on belief directions (such as inference-time intervention, Li et al., 2023) might
> > > be of help in these settings, it will be useful to first investigate what impact context has on belief
> > > directions. Our experiments investigate the behaviour of belief probes on sentences that appear
> > > in contexts with other related sentences. This enables us to determine how inferential contexts
> > > influence an LLM’s assessment of truth-values.

---

### Meta-Review · Area_Chair_7BbD · 2024-12-21

**Metareview:**

This paper concerns the degree to which linear representations of concepts such as truthfulness are universal, and to what degree they are influenced by contextual signals in the text.

In my view, the main point of discussion in the reviews is that it's somewhat unclear exactly what the problem being addressed here is. This manifests as reviewers questioning the motivation and experimental design. I quote the final comment from reviewer BKPZ as illustrative of the issue here:

"I agree that the main question—whether belief directions depend on context—is crucial. When reading this paper, I was hoping to see an analysis of how the estimated belief directions vary across different types of contexts, possibly inspired by the approach taken by Marks and Tegmark. For instance, it would be fascinating to examine whether there is any correlation between belief directions estimated from subsets of the data and distinct linear representations of other concepts, such as negation, structural elements, or topics.

However, the paper primarily focuses on estimating belief directions using larger text datasets and comparing them to the existing belief directions. Given that earlier contexts can alter the truth of sentences, it seems natural to include such context-sensitive data in the analysis. The central argument appears to be that larger datasets yield better belief directions, but I think it would be compelling to explore experiments with probes trained on different subsets of the dataset to better understand these dynamics."

In my view, if the reviewer has correctly understood the motivation and problem then the described experiment is indeed necessary. If the reviewer has fundamentally misunderstood the motivation and problem, then the paper needs a significant revision to clarify exactly what is happening here. (And, unclarity about the motivation was the chief complaint of fqrq).

Overall, my sense is that this paper will eventually be very strong, but it would benefit from a substantial revision before it is published.

**Additional Comments On Reviewer Discussion:**

see above

---

### Decision · Program_Chairs · 2025-01-22

Reject